# Towards Understanding Generalization via Decomposing Excess Risk Dynamics

**Jiaye Teng**[1,*]**, Jianhao Ma**[2,*]**, Yang Yuan**[1,3]
[1]Institute for Interdisciplinary Information Sciences, Tsinghua University
[2]Department of Industrial and Operational Engineering, University of Michigan, Ann Arbor
[3]Shanghai Qi Zhi Institute
`tjy20@mails.tsinghua.edu, jianhao@umich.edu`
`yuanyang@tsinghua.edu`

## Abstract

Generalization is one of the fundamental issues in machine learning. However, traditional techniques like uniform convergence may be unable to explain generalization under overparameterization (Nagarajan & Kolter, 2019). As alternative approaches, techniques based on *stability* analyze the training dynamics and derive algorithm-dependent generalization bounds. Unfortunately, the stability-based bounds are still far from explaining the surprising generalization in deep learning since neural networks usually suffer from unsatisfactory stability. This paper proposes a novel decomposition framework to improve the stability-based bounds via a more fine-grained analysis of the signal and noise, inspired by the observation that neural networks converge relatively slowly when fitting noise (which indicates better stability). Concretely, we decompose the excess risk dynamics and apply the stability-based bound only on the noise component. The decomposition framework performs well in both linear regimes (overparameterized linear regression) and non-linear regimes (diagonal matrix recovery). Experiments on neural networks verify the utility of the decomposition framework.

## 1 Introduction

Generalization is one of the essential mysteries uncovered in modern machine learning (Neyshabur et al., 2014; Zhang et al., 2016; Kawaguchi et al., 2017), measuring how the trained model performs on unseen data. One of the most popular approaches to generalization is *uniform convergence* (Mohri et al., 2018), which takes the supremum over parameter space to decouple the dependency between the training set and the trained model. However, Nagarajan & Kolter (2019) pointed out that uniform convergence itself might not be powerful enough to explain generalization, because the uniform bound can still be vacuous under overparameterized linear regimes.

One alternative solution beyond uniform convergence is to analyze the *generalization dynamics*, which measures the generalization gap during the training dynamics. *Stability*-based bound is among the most popular techniques in generalization dynamics analysis (Lei & Ying, 2020), which is derived from algorithmic stability (Bousquet & Elisseeff, 2002). Fortunately, one can derive non-vacuous bounds under general convex regimes using stability frameworks (Hardt et al., 2016). However, stability is still far from explaining the remarkable generalization abilities of neural networks, mainly due to two obstructions. Firstly, stability-based bound depends heavily on the gradient norm in non-convex regimes (Li et al., 2019), which is typically large at the beginning phase in training neural networks. Secondly, stability-based bound usually does not work well under general non-convex regimes (Hardt et al., 2016; Charles & Papailiopoulos, 2018; Zhou et al., 2018b) but neural networks are usually highly non-convex.

The aforementioned two obstructions mainly stem from the coarse-grained analysis of the signal and noise. As Zhang et al. (2016) argued, neural networks converge fast when fitting signal but converge

---

*Equal contribution.

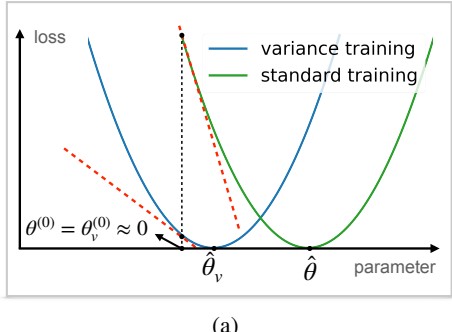 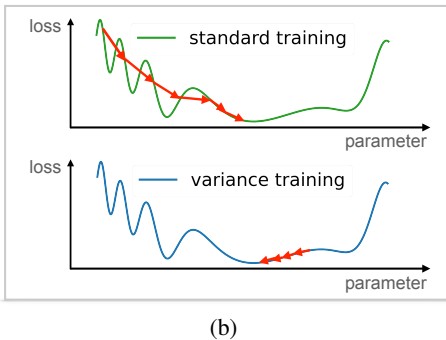

(a)                            (b)

Figure 1: (a) The gradient norm of standard training (training over noisy data) is larger than that of variance training (training over pure noise) at initialization $\boldsymbol{\theta}^{(0)} = \boldsymbol{\theta}_v^{(0)} \approx \mathbf{0}$. We denote the minimizers of the training loss by $\hat{\boldsymbol{\theta}}$, $\hat{\boldsymbol{\theta}}_v$, respectively (where the $y$-axis is the training loss). (b) The whole training loss landscape is highly non-convex while the trajectory of the variance training lies in a convex region due to the slow convergence. We defer the details to Appendix B.

relatively slowly when fitting noise[1], indicating that the training dynamics over signal and noise are significantly different. Consequently, on the one hand, the fast convergence of signal-related training contributes to a large gradient norm at the beginning phase (see Figure 1a), resulting in poor stability. On the other hand, the training on signal forces the trained parameter away from the initialization, making the whole training path highly non-convex (see Figure 1b). The above two phenomena inspire us to decompose the training dynamics into *noise* and *signal* components and *only* apply the stability-based analysis over the noise component. To demonstrate that such decomposition generally holds in practice, we conduct several experiments of neural networks on both synthetic and real-world datasets (see Figure 2 for more details).

Based on the above discussion, we improve the stability-based analysis by proposing a decomposition framework on *excess risk dynamics*[2], where we handle the noise and signal components separately via a bias-variance decomposition. In detail, we decompose the excess risk into variance excess risk (VER) and bias excess risk (BER), where VER measures how the model fits noise and BER measures how the model fits signal. Under the decomposition, we apply the stability-based techniques to VER and apply uniform convergence to BER inspired by Negrea et al. (2020). The decomposition framework accords with the theoretical and experimental evidence surprisingly well, providing that it outperforms stability-based bounds in both linear (overparameterized linear regression) and non-linear (diagonal matrix recovery) regimes.

We summarize our contributions as follows:

- We propose a new framework aiming at improving the traditional stability-based bounds, which is a novel approach to generalization dynamics analysis focusing on decomposing the excess risk dynamics into variance component and bias component. Starting from the overparameterized linear regression regimes, we show how to deploy the decomposition framework in practice, and the proposed framework outperforms the stability-based bounds.

- We theoretically analyze the excess risk decomposition beyond linear regimes. As a case study, we derive a generalization bound under diagonal matrix recovery regimes. To our best knowledge, this is the first work to analyze the generalization performance of diagonal matrix recovery.

- We conduct several experiments on both synthetic datasets and real-world datasets (MINIST, CIFAR-10) to validate the utility of the decomposition framework, indicating that the framework provides interesting insights for the generalization community.

---

[1]In this paper, we refer the signal to the clean data without the output noise, and the noise to the output noise. See Section 2 for the formal definitions.

[2]We decompose the excess risk, which is closely related to generalization, purely due to technical reasons. The excess risk dynamics tracks the excess risk during the training process.

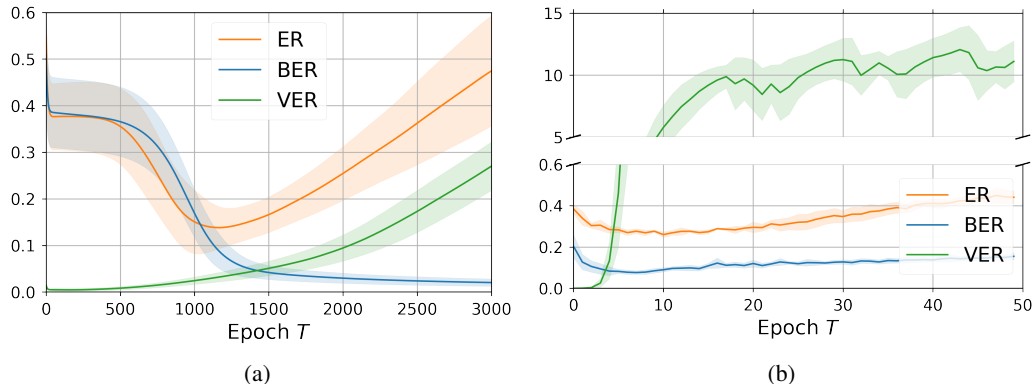

(a)                                           (b)

Figure 2: Experiments of neural networks on synthetic (linear ground truth, 3-layer NN, SGD) and real-world (MNIST, 3-layer CNN, Adam) datasets. The trend of excess risk dynamics (blue) meets its bias component (BER) at the beginning phase and meets its variance component (VER) afterwards, indicating that ER can indeed be decomposed into VER and BER. See Appendix A for more details.

## 1.1 RELATED WORK

**Stability-based Generalization.** The stability-based researches can be roughly split into two branches. One branch is about how algorithmic stability leads to generalization (Feldman & Vondrak, 2018; 2019; Bousquet et al., 2020). Another branch focus on how to calculate the stability parameter for specific problems, e.g., Hardt et al. (2016) prove a generalization bound scales linearly with time in convex regimes. Furthermore, researchers try to apply stability techniques into more general settings, e.g., non-smooth loss (Lei & Ying, 2020; Bassily et al., 2020), noisy gradient descent in non-convex regimes (Mou et al., 2018; Li et al., 2019) and stochastic gradient descent in non-convex regimes (Zhou et al., 2018b; Charles & Papailiopoulos, 2018; Zhang et al., 2021). In this paper, we mainly focus on applying the decomposition framework to improve the stability-based bound.

**Uniform Convergence** is widely used in generalization analysis. For bounded losses, the generalization gap is tightly bounded by its Rademacher Complexity (Koltchinskii & Panchenko, 2000; Koltchinskii, 2001; Koltchinskii et al., 2006). Furthermore, we reach faster rates under realizable assumption (Srebro et al., 2010). A line of work focuses on uniform convergence under neural network regimes, which is usually related to the parameter norm (Bartlett et al., 2017; Wei & Ma, 2019). However, as Nagarajan & Kolter (2019) pointed out, uniform convergence may be unable to explain generalization. Therefore, more techniques are explored to go beyond uniform convergence.

**Other Approaches to Generalization.** There are some other approaches to generalization, including PAC-Bayes (Neyshabur et al., 2017a; Dziugaite & Roy, 2017; Neyshabur et al., 2017b; Dziugaite & Roy, 2018; Zhou et al., 2018a; Yang et al., 2019), information-based bound (Russo & Zou, 2016; Xu & Raginsky, 2017; Banerjee & Montúfar, 2021; Haghifam et al., 2020; Steinke & Zakynthinou, 2020), and compression-based bound (Arora et al., 2018; Allen-Zhu et al., 2018; Arora et al., 2019).

**Bias-Variance Decomposition**. Bias-variance decomposition plays an important role in statistical analysis (Lehmann & Casella, 2006; Casella & Berger, 2021; Geman et al., 1992). Generally, high bias indicates that the model has poor predicting ability on average and high variance indicates that the model performs unstably. Bias-variance decomposition is widely used in machine learning analysis, e.g., adversarial training (Yu et al., 2021), double descent (Adlam & Pennington, 2020), uncertainty (Hu et al., 2020). Additionally, (Nakkiran et al., 2020) propose a decomposition framework from an online perspective relating the different optimization speeds on empirical and population loss. Oymak et al. (2019) applied bias-variance decomposition on the Jacobian of neural networks to explain their different performances on clean and noisy data. This paper considers a slightly different bias-variance decomposition following the analysis of SGD (Dieuleveut et al., 2016; Jain et al., 2018; Zou et al., 2021), focusing on the decomposition of the noisy output.

**Matrix Recovery.** Earlier methods for solving matrix recovery problems rely on the convex relaxation techniques for minimum norm solutions (Recht et al., 2010; Chandrasekaran et al., 2011). Recently, a branch of works focuses on the matrix factorization techniques with simple local search methods. It has been shown that there is no spurious local minima in the exact regime (Ge et al., 2016; 2017; Zhang et al., 2019). In the overparameterized regimes, it was first conjectured by Gunasekar et al. (2018) and then answered by Li et al. (2018) that gradient descent methods converge to the low-rank solution efficiently. Later, Zhuo et al. (2021) extends the conclusion to the noisy settings.

## 2 PRELIMINARY

In this section, we introduce necessary definitions, assumptions, and formally introduce previous techniques in dealing with generalization.

**Data distribution.** Let $\boldsymbol{x} \in \mathcal{X} \subset \mathbb{R}^p$ be the input and $\boldsymbol{y} \in \mathcal{Y} \subset \mathbb{R}$ be the output, where $\boldsymbol{x}, \boldsymbol{y}$ are generated from a joint distribution $(\boldsymbol{x}, \boldsymbol{y}) \sim \mathcal{P}$. Define $\mathcal{P}_{\boldsymbol{x}}, \mathcal{P}_{\boldsymbol{y}}$, and $\mathcal{P}_{\boldsymbol{y}|\boldsymbol{x}}$ as the corresponding marginal and conditional distributions, respectively. Given $n$ training samples $\mathcal{D} \triangleq \{\boldsymbol{x}_i, \boldsymbol{y}_i\}_{i \in [n]}$ generated from distribution $\mathcal{P}$, we denote its empirical distribution by $\mathcal{P}^n$. To simplify the notations, we define $X \in \mathbb{R}^{n \times p}$ as the design matrix and $Y \in \mathbb{R}^n$ as the response vector.

**Excess Risk.** Given the loss function $\ell(\boldsymbol{\theta}; \boldsymbol{x}, \boldsymbol{y})$ with parameter $\boldsymbol{\theta}$ and sample $(\boldsymbol{x}, \boldsymbol{y})$, we define the population loss as $\mathcal{L}(\boldsymbol{\theta}; \mathcal{P}) \triangleq \mathbb{E}_{(\boldsymbol{x}, \boldsymbol{y}) \sim \mathcal{P}} [\ell(\boldsymbol{\theta}; \boldsymbol{x}, \boldsymbol{y})]$ and its corresponding training loss as $\mathcal{L}(\boldsymbol{\theta}; \mathcal{P}^n) \triangleq \frac{1}{n} \sum_i \ell(\boldsymbol{\theta}; \boldsymbol{x}_i, \boldsymbol{y}_i)$. Let $\mathcal{A}_t$ denote the optimization algorithm which takes the dataset $\mathcal{D}$ as an input and returns the trained parameter $\hat{\boldsymbol{\theta}}^{(t)}$ at time $t$, namely, $\mathcal{A}_t(\mathcal{D}) = \hat{\boldsymbol{\theta}}^{(t)}$. During the analysis, we focus on the *excess risk dynamics* $\mathcal{E}_{\mathcal{L}}(\hat{\boldsymbol{\theta}}^{(t)}; \mathcal{P})$, which measures how the trained parameter $\hat{\boldsymbol{\theta}}^{(t)}$ performs on the population loss:

$$\mathcal{E}_{\mathcal{L}}(\hat{\boldsymbol{\theta}}^{(t)}; \mathcal{P}) \triangleq \mathcal{L}(\hat{\boldsymbol{\theta}}^{(t)}; \mathcal{P}) - \min_{\boldsymbol{\theta}} \mathcal{L}(\boldsymbol{\theta}; \mathcal{P}).$$

Although the minimizer may not be unique, we define $\mathcal{L}(\boldsymbol{\theta}^*; \mathcal{P}) \triangleq \min_{\boldsymbol{\theta}} \mathcal{L}(\boldsymbol{\theta}; \mathcal{P})$ where $\boldsymbol{\theta}^*$ denotes arbitrarily one of the minimizers. Additionally, bounding the generalization gap $\mathcal{L}(\hat{\boldsymbol{\theta}}^{(t)}; \mathcal{P}) - \mathcal{L}(\hat{\boldsymbol{\theta}}^{(t)}; \mathcal{P}^n)$ suffices to bound the excess risk under *Empirical Risk Minimization* (ERM) framework by Equation equation 1. Therefore, we mainly discuss the excess risk following Bartlett et al. (2020).

$$\mathcal{E}_{\mathcal{L}}(\hat{\boldsymbol{\theta}}^{(t)}; \mathcal{P}) = \underbrace{\left[\mathcal{L}(\hat{\boldsymbol{\theta}}^{(t)}; \mathcal{P}) - \mathcal{L}(\hat{\boldsymbol{\theta}}^{(t)}; \mathcal{P}^n)\right]}_{\text{generalization gap}} + \underbrace{\left[\mathcal{L}(\hat{\boldsymbol{\theta}}^{(t)}; \mathcal{P}^n) - \mathcal{L}(\boldsymbol{\theta}^*; \mathcal{P}^n)\right]}_{\leq 0 \text{ under ERM}} + \underbrace{\left[\mathcal{L}(\boldsymbol{\theta}^*; \mathcal{P}^n) - \mathcal{L}(\boldsymbol{\theta}^*; \mathcal{P})\right]}_{\approx 0 \text{ by concentration inequalities}}.$$

$$(1)$$

**VER and BER.** As discussed in Section 1, we aim to decompose the excess risk into the *variance excess risk (VER)* and the *bias excess risk (BER)* defined in Definition 1. The decomposition focuses on the noisy output regimes and we split the output $\boldsymbol{y}$ into the signal component $\mathbb{E}[\boldsymbol{y}|\boldsymbol{x}]$ and the noise component $\boldsymbol{y} - \mathbb{E}[\boldsymbol{y}|\boldsymbol{x}]$. Informally, VER measures how the model performs on pure noise, and BER measures how the model performs on clean data.

**Definition 1 (VER and BER)** *Given $(\boldsymbol{x}, \boldsymbol{y}) \sim \mathcal{P}$, let $(\boldsymbol{x}, \mathbb{E}[\boldsymbol{y}|\boldsymbol{x}]) \sim \mathcal{P}_b$ denote the signal distribution and $(\boldsymbol{x}, \boldsymbol{y} - \mathbb{E}[\boldsymbol{y}|\boldsymbol{x}]) \sim \mathcal{P}_v$ denote the noise distribution. The variance excess risk (VER) $\mathcal{E}_{\mathcal{L}}^v(\boldsymbol{\theta}; \mathcal{P})$ and bias excess risk (BER) $\mathcal{E}_{\mathcal{L}}^b(\boldsymbol{\theta}; \mathcal{P})$ are defined as:*

$$\mathcal{E}_{\mathcal{L}}^v(\boldsymbol{\theta}; \mathcal{P}) \triangleq \mathcal{E}_{\mathcal{L}}(\boldsymbol{\theta}; \mathcal{P}_v), \quad \mathcal{E}_{\mathcal{L}}^b(\boldsymbol{\theta}; \mathcal{P}) \triangleq \mathcal{E}_{\mathcal{L}}(\boldsymbol{\theta}; \mathcal{P}_b).$$

To better illustrate VER and BER, we consider three surrogate training dynamics: Standard Training, Variance Training, and Bias Training, corresponding to ER, VER, and BER, respectively.

Standard Training: Training process over the noisy data $(X, Y)$ from the initialization $\boldsymbol{\theta}^{(0)}$. We denote the trained parameter at time $t$ by $\hat{\boldsymbol{\theta}}^{(t)}$.

Variance Training: Training process over the pure noise $(X, Y - \mathbb{E}[Y|X])$ from the initialization $\boldsymbol{\theta}_v^{(0)}$. We denote the trained parameter at time $t$ by $\hat{\boldsymbol{\theta}}_v^{(t)}$.

Bias Training: Training process over the clean data $(X, \mathbb{E}[Y|X])$ from the initialization $\boldsymbol{\theta}_b^{(0)}$. We

denote the trained parameter at time $t$ by $\hat{\boldsymbol{\theta}}_b^{(t)}$.

When the context is clear, although the trained parameters $\hat{\boldsymbol{\theta}}^{(t)}, \hat{\boldsymbol{\theta}}_v^{(t)}, \hat{\boldsymbol{\theta}}_b^{(t)}$ are related to the corresponding initialization and the algorithms, we omit the dependency. Besides, we denote $\boldsymbol{\theta}^*, \boldsymbol{\theta}_v^*$ and $\boldsymbol{\theta}_b^*$ the optimal parameters which minimize the corresponding population loss $\mathcal{L}(\boldsymbol{\theta}; \mathcal{P})$, $\mathcal{L}(\boldsymbol{\theta}; \mathcal{P}_v)$, and $\mathcal{L}(\boldsymbol{\theta}; \mathcal{P}_b)$, respectively.

**Techniques in generalization.** We next introduce two techniques in generalization analysis including *stability*-based techniques in Proposition 1 and uniform convergence in Proposition 2. These techniques will be revisited in Section 3.

**Proposition 1 (Stability Bound from Feldman & Vondrak (2019))** *Assume that algorithm $\mathcal{A}_t$ is $\epsilon$-uniformly-stable at time $t$, meaning that for any two dataset $\mathcal{D}$ and $\mathcal{D}'$ with only one different data point, we have*

$$\sup_{(\boldsymbol{x}, \boldsymbol{y})} \mathbb{E}_{\mathcal{A}} \left[ \ell(\mathcal{A}_t(\mathcal{D}); \boldsymbol{x}, \boldsymbol{y}) - \ell(\mathcal{A}_t(\mathcal{D}'); \boldsymbol{x}, \boldsymbol{y}) \right] \leq \epsilon.$$

*Then the following inequality holds[3] with probability at least $1 - \delta$:*

$$\left| \mathcal{L}\left(\mathcal{A}_t(\mathcal{D}); \mathcal{P}\right) - \mathcal{L}\left(\mathcal{A}_t(\mathcal{D}); \mathcal{P}^n\right) \right| = \mathcal{O}\left( \epsilon \log(n) \log(n/\delta) + \sqrt{\frac{\log(1/\delta)}{n}} \right).$$

**Proposition 2 (Uniform Convergence from Wainwright (2019))** *Uniform convergence decouples the dependency between the trained parameter and the training set by taking supremum over a parameter space that is independent of training data, namely*

$$\mathcal{L}(\mathcal{A}_t(\mathcal{D}); \mathcal{P}) - \mathcal{L}(\mathcal{A}_t(\mathcal{D}); \mathcal{P}^n) \leq \sup_{\boldsymbol{\theta} \in \mathcal{B}} \left[ \mathcal{L}(\boldsymbol{\theta}; \mathcal{P}) - \mathcal{L}(\boldsymbol{\theta}; \mathcal{P}^n) \right].$$

*where $\mathcal{B}$ is independent of dataset $\mathcal{D}$ and $\mathcal{A}_t(\mathcal{D}) \in \mathcal{B}$ for any time $t$.*

## 3 WARM-UP: DECOMPOSITION UNDER LINEAR REGIMES

In this section, we consider the overparameterized linear regression with linear assumptions, where the sample size is less than the data dimension, namely, $n < p$. We will show how the decomposition framework works and the improvements compared to traditional stability-based bounds. Before diving into the details, we emphasize the importance of studying overparameterized linear regression. As the arguably simplest models, we expect a framework can at least work under such regimes. Besides, recent works (e.g., Jacot et al. (2018)) show that neural networks converge to overparametrized linear models (with respect to the parameters) as the width goes to infinity under some regularity conditions. Therefore, studying overparameterized linear regression provides enormous intuition to neural networks.

**Settings.** When the context is clear, we reuse the notations in Section 2. Without loss of generality, we assume that $\mathbb{E}[\boldsymbol{x}] = \boldsymbol{0}$. Besides, we assume a linear ground truth $\boldsymbol{y} = \boldsymbol{x}^\top \boldsymbol{\theta}^* + \epsilon$ where $\mathbb{E}[\epsilon | \boldsymbol{x}] = 0$. Let $\Sigma_{\boldsymbol{x}} \triangleq \mathbb{E}[\boldsymbol{x}\boldsymbol{x}^\top]$ denote the covariance matrix. Given $\ell_2$ loss $\ell(\boldsymbol{\theta}; \boldsymbol{x}, \boldsymbol{y}) = (\boldsymbol{y} - \boldsymbol{x}^\top \boldsymbol{\theta})^2$, we apply gradient descent (GD) with constant step size $\lambda$ and initialization $\boldsymbol{\theta}^{(0)} = \boldsymbol{0}$, namely, $\hat{\boldsymbol{\theta}}^{(t+1)} = \hat{\boldsymbol{\theta}}^{(t)} + \frac{\lambda}{n} \sum_i \boldsymbol{x}_i(\boldsymbol{y}_i - \boldsymbol{x}_i^\top \hat{\boldsymbol{\theta}}^{(t)})$. We provide the main results of overparameterized linear regression in Theorem 1 and then introduce how to apply the decomposition framework.

**Theorem 1 (Linear Regression Regimes)** *Under the overparameterized linear regression settings, assume that $w = \frac{\boldsymbol{\theta}^{*,\top} \boldsymbol{x}}{\sqrt{\boldsymbol{\theta}^{*,\top} \Sigma_{\boldsymbol{x}} \boldsymbol{\theta}^*}}$ is $\sigma_w^2$-subgaussian. If $\|\boldsymbol{x}\| \leq 1$, $|\epsilon| \leq V$, $\sup_{t \in [T]} \|\hat{\boldsymbol{\theta}}_v^{(t)}\| \leq B$ are all bounded, the following inequality holds with probability at least $1 - \delta$:*

$$\mathcal{E}_{\mathcal{L}}(\hat{\boldsymbol{\theta}}^{(T)}; \mathcal{P}) = \tilde{\mathcal{O}}\left( \max\left\{ 1, \boldsymbol{\theta}^{*,\top} \Sigma_{\boldsymbol{x}} \boldsymbol{\theta}^* \sigma_w^2, [V + B]^2 \right\} \sqrt{\frac{\log(4/\delta)}{n}} + \frac{\|\boldsymbol{\theta}^*\|^2}{\lambda T} + \frac{T\lambda[V + B]^2}{n} \right),$$

*where the probability is taken over the randomness of training data.*

---

[3]In the rest of the paper, we use $\mathcal{O}$ or $\lesssim$ to omit the constant dependency and use $\tilde{\mathcal{O}}$ to omit the log dependency of $n$. For example, we write $2/n = \mathcal{O}(1/n)$ or $2/n \lesssim 1/n$, $\log(n)/n = \tilde{\mathcal{O}}(1/n)$, and $\log^2(n)/n = \tilde{\mathcal{O}}(1/n)$.

By setting $T = \boldsymbol{\theta}(\sqrt{n})$ in Theorem 1, the upper bound is approximately $\tilde{\mathcal{O}}(\frac{1}{\sqrt{n}})$, indicating that the estimator at time $T$ is consistent[4]. Below we show the comparison with some existing bounds, including benign overfitting bound and traditional stability-based bound[5].

Comparison with benign overfitting. The bound in Bartlett et al. (2020) mainly focus on the min-norm solution (as $T \to \infty$, the parameter trained in GD converges to the min-norm solution). Instead, the bound in Theorem 1 does not require the min-norm solution assumption.

Comparison with stability-based bound. The stability-based bound (without applying the decomposition framework) is $\tilde{\mathcal{O}}\left(\max\{1, [V + B']^2\}\sqrt{\frac{\log(2/\delta)}{2n}} + \frac{T\lambda}{n}[V + B']^2\right)$, where $B' = \sup_t \|\hat{\boldsymbol{\theta}}^{(t)}\|$ denotes the bound of the trained parameter in standard training. As a comparison, the notation $B$ in Theorem 1 denotes the bound of the trained parameter in *variance training*. One major difference between $B$ and $B'$ is that: $B$ is independent of $\|\boldsymbol{\theta}^*\|$ while $B'$ is closely related to $\|\boldsymbol{\theta}^*\|$. Therefore, with a large time $T$, the bound in Theorem 1 outperforms the stability-based bound when $\|\boldsymbol{\theta}^*\|$ is large compared to $V$ (namely, large signal-to-noise ratio)[6]. We refer to Appendix F.1 for more discussion.

**Proof Sketch.** We defer the proof details to Appendix C due to space limitations. Firstly, we provide Lemma 1 to show that the excess risk can be decomposed into VER and BER.

**Lemma 1** *Under the overparameterized linear regression settings with initialization $\boldsymbol{\theta}^{(0)} = \boldsymbol{\theta}_b^{(0)} = \boldsymbol{\theta}_v^{(0)} = \mathbf{0}$, for any time $T$, the following decomposition inequality holds:*

$$\mathcal{E}_{\mathcal{L}}(\hat{\boldsymbol{\theta}}^{(T)}; \mathcal{P}) \leq 2\mathcal{E}_{\mathcal{L}}^v(\hat{\boldsymbol{\theta}}_v^{(T)}; \mathcal{P}) + 2\mathcal{E}_{\mathcal{L}}^b(\hat{\boldsymbol{\theta}}_b^{(T)}; \mathcal{P}).$$

We next bound VER and BER separately in Lemma 2 and Lemma 3. When applying the stability-based bound on VER, we first convert VER into a generalization gap (see Equation 1) and then bound three terms individually.

**Lemma 2** *Under the overparameterized linear regression settings, assume that $\|\boldsymbol{x}\| \leq 1$, $|\epsilon| \leq V$, $\sup_t \|\hat{\boldsymbol{\theta}}_v^{(t)}\| \leq B$ are all bounded. Consider the gradient descent algorithm with constant stepsize $\lambda$, with probability at least $1 - \delta$:*

$$\mathcal{E}_{\mathcal{L}}^v\left(\hat{\boldsymbol{\theta}}_v^{(T)}; \mathcal{P}\right) = \mathcal{O}\left(\frac{T\lambda}{n}[V + B]^2 \log(n)\log(2n/\delta) + \max\left\{1, [V + B]^2\right\}\sqrt{\frac{\log(2/\delta)}{2n}}\right). \quad (2)$$

In Lemma 3, we apply uniform convergence to bound BER. Although we can bound it by directly calculating the excess risk, we still apply uniform convergence, because exact calculating is limited to a specific problem while uniform convergence is much more general. To link the uniform convergence and excess risk, we split the excess risk into two pieces inspired by Negrea et al. (2020). For the first piece, which stands for the generalization gap, we bound it by uniform convergence. For the second piece, which stands for the (surrogate) training loss, we derive a bound decreasing with time $T$.

**Lemma 3** *Under the overparameterized linear regression settings, assume that $w = \frac{\boldsymbol{\theta}^{*,\top}\boldsymbol{x}}{\sqrt{\boldsymbol{\theta}^{*,\top}\Sigma_{\boldsymbol{x}}\boldsymbol{\theta}^*}}$ is $\sigma_w^2$-subgaussian, then with probability at least $1 - \delta$:*

$$\mathcal{E}_{\mathcal{L}}^b(\hat{\boldsymbol{\theta}}_b^{(T)}; \mathcal{P}) = \mathcal{O}\left(\frac{\boldsymbol{\theta}^{*,\top}\Sigma_{\boldsymbol{x}}\boldsymbol{\theta}^*}{\sqrt{n}}\sigma_w^2\sqrt{\log\left(\frac{2}{\delta}\right)} + \frac{1}{\lambda[2T + 1]}\|\boldsymbol{\theta}^*\|^2\right). \quad (3)$$

Although we do not present it explicitly, the results in Lemma 3 are derived under uniform convergence frameworks. Combining Lemma 1, Lemma 2 and Lemma 3 leads to Theorem 1. $\qquad\square$

---

[4]Consistency means the upper bound converges to zero as the number of training samples goes to infinity.

[5]We remark that the assumptions in Theorem 1 can be relaxed. Firstly, assuming $\boldsymbol{x}$ is subGaussian suffices to show that $w$ is subGaussian. Secondly, assuming $\epsilon$ is subGaussian suffices to bound $|\epsilon| \leq V$ with a $\log(n)$ cost.

[6]A large time $t$ can guarantee that the dominating term in Theorem 1 is $\frac{T\lambda}{n}[V + B']^2$. We remark that with a large signal-to-noise ratio, we have $\frac{T\lambda}{n}[V + B']^2 \geq \frac{T\lambda}{n}[V + B]^2$ ).

## 4 DECOMPOSITION UNDER GENERAL REGIMES

This section mainly discusses how to decompose the excess risk into variance component (VER) and bias component (BER) beyond linear regimes. Due to a fine-grained analysis on the signal and noise, the decomposition framework improves the stability-based bounds under the general regimes. We emphasize that the methods derived in this section are nevertheless the only way to reach the decomposition goal and we leave more explorations for future work. To simplify the discussion, we assume the minimizer $\boldsymbol{\theta}^*$ of the excess risk exists and is unique. Before diving into the main theorem, it is necessary to introduce the definition of *local sharpness* for the excess risk function, measuring the smoothness of the excess risk function around the minimizer $\boldsymbol{\theta}^*$.

**Definition 2 (Local Sharpness)** *For a given excess risk function $\mathcal{E}_{\mathcal{L}}(\boldsymbol{\theta}; \mathcal{P})$ with $\mathcal{E}_{\mathcal{L}}(\boldsymbol{\theta}^*; \mathcal{P}) = 0$, we define $s > 0$ as the local sharpness factor when the following equation holds[7]:*

$$0 < \lim_{\boldsymbol{\theta}: \|\boldsymbol{\theta} - \boldsymbol{\theta}^*\| \to 0} \frac{\mathcal{E}_{\mathcal{L}}(\boldsymbol{\theta}; \mathcal{P})}{\|\boldsymbol{\theta} - \boldsymbol{\theta}^*\|^s} < \infty.$$

**Assumption 1 (Uniform Bound)** *Assume that for excess risk function $\mathcal{E}_{\mathcal{L}}(\boldsymbol{\theta}; \mathcal{P})$ with local sharpness $s$, given $\mathcal{P}_{\boldsymbol{x}}$, for any $\mathcal{P}_{\boldsymbol{y}|\boldsymbol{x}}$ and $\boldsymbol{\theta}$, the uniform bounds $0 < m_u(\mathcal{P}_{\boldsymbol{x}}) \le M_u(\mathcal{P}_{\boldsymbol{x}})$ exist:*

$$m_u(\mathcal{P}_{\boldsymbol{x}}) \le \frac{\mathcal{E}_{\mathcal{L}}(\boldsymbol{\theta}; \mathcal{P})}{\|\boldsymbol{\theta} - \boldsymbol{\theta}^*\|^s} \le M_u(\mathcal{P}_{\boldsymbol{x}}).$$

*When the context is clear, we omit the dependency of $\mathcal{P}_{\boldsymbol{x}}$ and denote the bounds by $m_u$ and $M_u$.*

Roughly speaking, the uniform bounds act as the maximal and minimal eigenvalues of the excess risk function $\mathcal{E}_{\mathcal{L}}(\boldsymbol{\theta}; \mathcal{P})$. For example, under the overparameterized linear regression regimes (Section 3), the uniform bounds are the maximum and minimum eigenvalues of the covariance matrix $\Sigma_{\boldsymbol{x}}$.

**Assumption 2 (Dynamics Decomposition Condition, DDC)** *We assume the trained parameters $\hat{\boldsymbol{\theta}}^{(T)}$ are $(a, C, C')$-bounded at time $T$, namely:*

$$\|\hat{\boldsymbol{\theta}}^{(T)} - \boldsymbol{\theta}^*\| \le a(\|\hat{\boldsymbol{\theta}}_v^{(T)} - \boldsymbol{\theta}_v^*\| + \|\hat{\boldsymbol{\theta}}_b^{(T)} - \boldsymbol{\theta}_b^*\|) + \frac{C}{\sqrt{T}} + \frac{C'}{\sqrt{n}}, \tag{4}$$

*where $n$ denotes the size of training samples, $T$ denotes the training time, and $\boldsymbol{\theta}^*$, $\boldsymbol{\theta}_v^*$ and $\boldsymbol{\theta}_b^*$ denote the minimizers of the corresponding population loss $\mathcal{L}(\boldsymbol{\theta}; \mathcal{P})$, $\mathcal{L}(\boldsymbol{\theta}; \mathcal{P}_v)$, and $\mathcal{L}(\boldsymbol{\theta}; \mathcal{P}_b)$, respectively.*

We emphasize that although we use $\frac{C}{\sqrt{T}} + \frac{C'}{\sqrt{n}}$ in Assumption 2, it can be replaced with any $o(1)$ term which converges to zero when $T, n$ go to infinity. Empirically, several experiments demonstrate the generality of DDC and its variety (see Figure 3). From the theoretical perspective, we derive that overparameterized linear regression is $(1, 0, 0)$-bounded in DDC, and diagonal matrix recovery regimes satisfy DDC (see Section 4.1). We next show how to transfer DDC to the Decomposition Theorem in Theorem 2.

**Theorem 2 (Decomposition Theorem)** *Assume that the excess risk has a unique minimizer with local sharpness factor $s$. Under Assumption 1 and Assumption 2, the following decomposition inequality holds:*

$$\mathcal{E}_{\mathcal{L}}(\hat{\boldsymbol{\theta}}^{(T)}; \mathcal{P}) \le [4a]^s \frac{M_u}{m_u} \left[\mathcal{E}_{\mathcal{L}}^v(\hat{\boldsymbol{\theta}}_v^{(T)}; \mathcal{P}) + \mathcal{E}_{\mathcal{L}}^b(\hat{\boldsymbol{\theta}}_b^{(T)}; \mathcal{P})\right] + M_u \left[\frac{4C}{\sqrt{T}}\right]^s + M_u \left[\frac{4C'}{\sqrt{n}}\right]^s. \tag{5}$$

We defer the proofs to Appendix D due to limited space. In Theorem 2, Assumption 1 and Assumption 2 focus on the population loss and the training dynamics individually. We remark that our proposed generalization bound is algorithm-dependent due to parameters $(a, C, C')$ in DDC.

By Theorem 2, we successfully decompose the excess risk dynamics into VER and BER. The remaining question is how to bound VER and BER individually. As we will derive in Section 4.1 via a case study on diagonal matrix recovery, we bound VER by stability techniques and BER by uniform convergence.

---

[7]Here $\| \cdot \|$ refers to the $\ell_2$-norm.

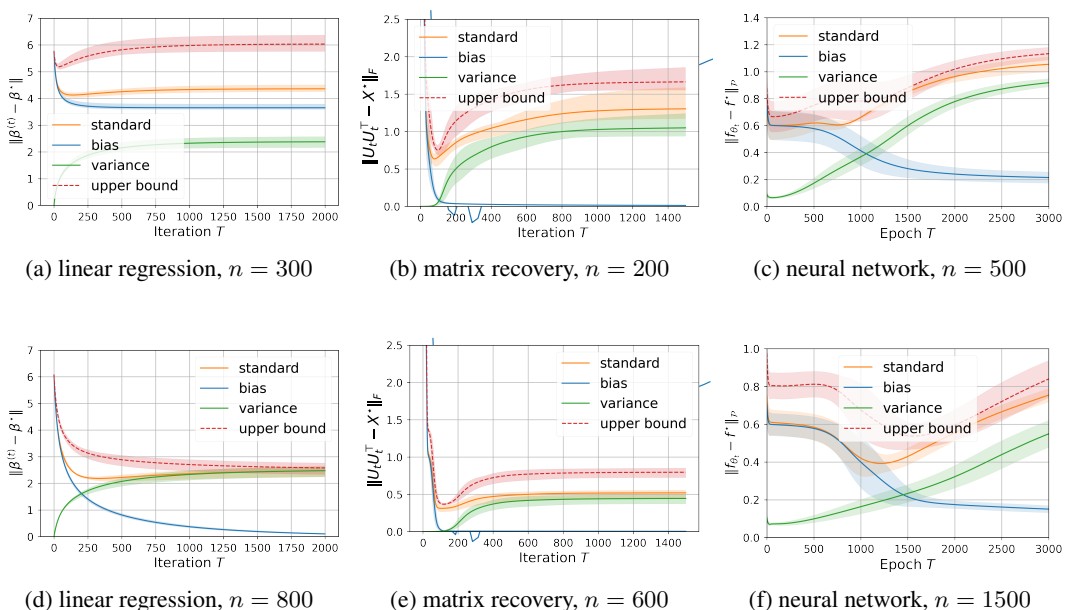

Figure 3: DDC holds in overparameterized linear regression, general matrix recovery, and neural networks. For linear regression and neural networks, we set $a = 1, C = C' = 0$ in DDC. For matrix recovery problems, we set $a = 1, C = 0, C' = 8.5$ in DDC. The upper bound (the right hand side in DDC, red line) is over the standard norm (the left hand side in DDC, orange line), indicating that DDC holds in general. We defer the experimental details to Appendix B.

## 4.1 DIAGONAL MATRIX RECOVERY

From the previous discussion, we now conclude the basic procedure when applying the decomposition framework under Theorem 2. Generally, there are three questions to be answered in a specific problem: (a) Does the problem satisfy DDC? (b) How to bound VER using stability? (c) How to bound BER using uniform convergence? To better illustrate how to solve the three questions individually, we provide a case study on diagonal matrix recovery, a special and simplified case of a general low-rank matrix recovery problem but still retains the key properties. The diagonal matrix recovery regime and its variants are studied as a prototypical problem in Gissin et al. (2019); HaoChen et al. (2021).

**Settings.** Let $X^\star = \text{Diag}\{\sigma_1^\star, \cdots, \sigma_r^\star, 0, \cdots, 0\} \in \mathbb{R}^{d \times d}$ denote a rank-$r$ (low rank) diagonal matrix, where $\sigma_1^\star \geq \cdots \geq \sigma_r^\star > 0$. We define $A \in \mathbb{R}^{d \times d}$ as the measurement matrix which is diagonal and $y \in \mathbb{R}^d$ as the measurement vector. Assume the ground truth $y_i = A_i \sigma_i^\star + \varepsilon_i$ where $y_i, A_i$ denotes their corresponding $i$-th element, $\epsilon_i$ denotes the noise, and $\sigma_i^\star = 0$ if $r < i \leq d$. Our goal is to recover $X^\star$ from dataset $\mathcal{D} = \{A^{(i)}, y^{(i)}\}_{i \in [n]}$ with $n$ training samples.

In the training process, we use the diagonal matrix $U = \text{Diag}\{u_1, \cdots, u_d\}$ to predict $X^\star$ via $\hat{X} = UU^\top$. Given the loss function[8] $\mathcal{L}_n(U) = \frac{1}{n} \sum_{j=1}^n \left\| y^{(j)} - A^{(j)} \odot UU^\top \right\|^2$, we train the model with Gradient Flow[9] $\dot{U}_t = -\nabla \mathcal{L}_n(U_t)$. During the analysis, we assume that $A_i$ is $\mathcal{O}(1)$-uniformly bounded with unit variance, and $\varepsilon_i$ is $\nu^2$-subGaussian with uniform bounds $|\varepsilon_i^{(j)}| \leq V$. We emphasize that the training procedure of diagonal matrix recovery is fundamentally different from linear regimes since $\mathcal{L}_n(U)$ is non-convex with respect to $U$.

**Theorem 3 (Decomposition Theorem in Diagonal Matrix Recovery)** *In the diagonal matrix recovery settings, if we set the initialization $U_0 = \alpha I$ where $\alpha > 0$ is the initialization scale, then for*

---

[8]We use $\odot$ to represent the Hadamard product.

[9]We use gradient flow instead of gradient descent mainly due to technical simplicity.

*any $0 < t < \infty$, with probability at least $1 - \delta$, we have:*

$$\mathcal{E}_{\mathcal{L}}(U_t; \mathcal{P}) \lesssim \left(\|X^\star\|_F^2 + dV^2 + d\alpha^4\right) \sqrt{\frac{\log(d/\delta)}{n}} + \sum_{j=1}^r \frac{\sigma_j^4}{\sigma_j^2 + \alpha^4 e^{\Omega(\sigma_j t)}} + \frac{\alpha^2 d}{t}$$

$$+ \frac{dV^2(t+1)}{n} \log(n) \log\left(\frac{2dn}{\delta}\right) + d\alpha^2 + \log^2\left(\frac{1}{\alpha^2}\right) \frac{r\nu^2 \log(r/\delta)}{n}. \tag{6}$$

By setting $\alpha = \Theta\left((d^2 n)^{-\frac{1}{4}}\right)$ and $t = \Theta\left(\log(dn\sigma_r)/\sigma_r\right)$ in Theorem 3, the upper bound is approximately $\tilde{\mathcal{O}}\left(\frac{dV^2 + \|X^\star\|_F^2}{\sqrt{n}}\right)$, indicating that the estimator at time $t$ is consistent.

Comparison with stability-based bound. The stability-based bound (without applying the decomposition framework) is $\tilde{\mathcal{O}}\left(\frac{dV^2 + \|X^\star\|_F^2}{\sqrt{n}} + \frac{(dV^2 + \|X^\star\|_\star)(t+1)}{n}\right)$. As a comparison, the bound in Theorem 3 outperforms the stability-based bound in the sense that we deduce the coefficient of the blowing-up term $\frac{t+1}{n}$ from $dV^2 + \|X^\star\|_\star$ to $dV^2$, which is a significant improvement with a large signal-to-noise ratio under large time $t$. Besides, the bound in Theorem 3 captures the bias-variance tradeoff, meaning that the excess risk curve falls first and then rises.

**Proof Sketch.** We defer the proof details to Appendix E due to space limitations. Similar to Section 3, the proof of Theorem 3 consists of the following three lemmas, corresponding to the DDC, BER, and VER terms separately. We first prove that diagonal matrix recovery regimes satisfy DDC in Lemma 4.

**Lemma 4** *Under the assumptions in Theorem 3, for any $0 \leq t < \infty$, with probability at least $1 - \delta$, we have*

$$\left\|U_t U_t^\top - X^\star\right\|_F \leq \left\|U_t^b U_t^{b,\top} - X^\star\right\|_F + \left\|U_t^v U_t^{v,\top}\right\|_F + \mathcal{O}\left(\log\left(\frac{1}{\alpha^2}\right)\sqrt{\frac{r\nu^2 \log(r/\delta)}{n}}\right). \tag{7}$$

Different from the overparameterized linear regression regimes, Lemma 4 suffers from a $\mathcal{O}\left(\frac{1}{\sqrt{n}}\right)$ factor due to the non-linearity. This phenomenon can be verified empirically in Figure 3b and Figure 3e. We next apply stability techniques and uniform convergence to tackle VER and BER in Lemma 5 and Lemma 6, respectively.

**Lemma 5** *For VER, under the assumptions in Theorem 3, with probability at least $1 - \delta$, we have*

$$\mathcal{E}_{\mathcal{L}}^v(U_t; \mathcal{P}) \lesssim \frac{dV^2(t+1)}{n} \log(n) \log\left(\frac{2dn}{\delta}\right) + d\alpha^2 + dV^2 \sqrt{\frac{\log(2d/\delta)}{n}}. \tag{8}$$

**Lemma 6** *For BER, under the assumptions in Theorem 3, with probability at least $1 - \delta$, we have*

$$\mathcal{E}_{\mathcal{L}}^b(U_t; \mathcal{P}) \lesssim \left(\|X^\star\|_F^2 + d\alpha^4\right) \sqrt{\frac{\log(d/\delta)}{n}} + \sum_{j=1}^r \frac{\sigma_j^4}{\sigma_j^2 + \alpha^4 e^{\Omega(\sigma_j t)}} + \frac{\alpha^2 d}{t}. \tag{9}$$

Combining the above lemmas leads to Theorem 3. $\qquad\square$

**Remark:** Directly applying Theorem 2 under overparameterized linear regression regimes costs a loss on condition number. The cost comes from the DDC condition where we transform the function space (excess risk function) to the parameter space (the distance between the trained parameter and the minimizer). We emphasize that the field of decomposition framework is indeed much larger than Theorem 2, and there are other different approaches beyond Theorem 2. For example, informally, by introducing a pre-conditioner matrix $A$, changing DDC from the form $\|\hat{\theta} - \theta^\star\|$ to $\|A\hat{\theta} - A\theta^\star\|$ and setting $A = \Sigma_x^{1/2}$, one can avoid the cost of the condition number. We leave the technical discussion for future work.

ACKNOWLEDGMENTS

This work has been partially supported by National Key R&D Program of China (2019AAA0105200) and 2030 innovation megaprojects of china (programme on new generation artificial intelligence) Grant No.2021AAA0150000. The authors would like to thank Ruiqi Gao for his insightful suggestions. We also thank Tianle Cai, Haowei He, Kaixuan Huang, and, Jingzhao Zhang for their helpful discussions.

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

# Supplementary Materials

In this section, we first provide additional numerical experiment in Section A and figure illustration in Section B. We then give all the omitted proofs of lemmas and theorems in Section C, Section D, and Section E. We finally provide some further discussions, including why we apply stability on VER, the motivation behind DDC and a relaxed version of DDC.

## A    NUMERICAL EXPERIMENT

In this section we provide additional numerical experiments on neural network to verify that our theory is satisfied for general neural network architectures and optimization algorithms.

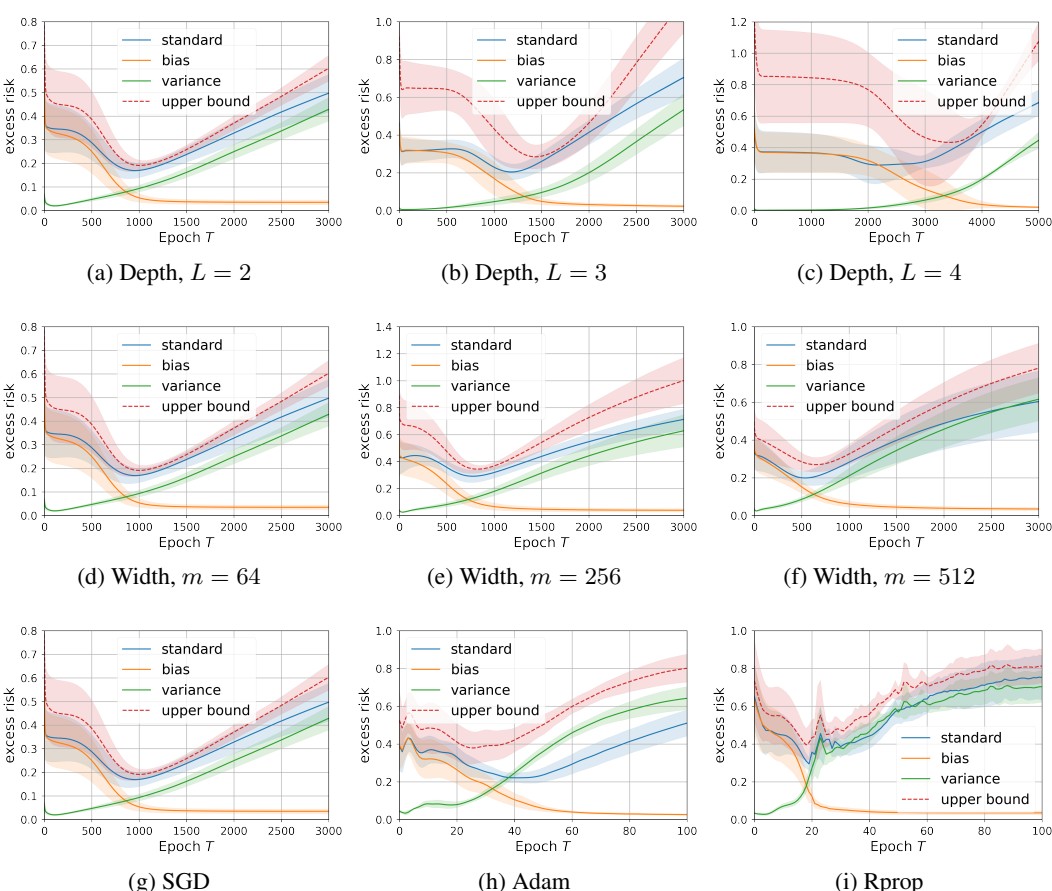

Figure 4: Numerical results in Section A.1.

## A.1    SYNTHETIC EXPERIMENT

In this section, our goal is to recover a sparse linear function $f^\star(\boldsymbol{x}) = \langle \boldsymbol{\theta}^\star, \boldsymbol{x} \rangle$ ($\boldsymbol{x} \in \mathbb{R}^{30}$) with fully connected ReLU network. For each setting, we run $5$ independent trials and report their average performances (the solid line) and the corresponding standard deviations (the error bar).

**Effects of depth:**    We first explore the effect of depth and width for the excess risk decomposition property. In this experiment, we fix the width of each layer to be $64$, and use SGD as the optimizer with Gaussian initialization ($\mathcal{N}(0, \sigma^2)$ where $\sigma = 1 \times 10^{-3}$) and stepsize $\eta = 1 \times 10^{-2}$. For the dash line, we plot $a(\mathsf{BER} + \mathsf{VER})$ to verify the performance of our risk decomposition framework. Here $a$ can be regarded as a measure of the success of our theory. Ideally $a$ should be a small constant so that

our bound is non-vacuous. Specific to this experiment, for Figure 4a, 4b, 4c, we set $a = 1.3, 2.0, 2.3$, respectively. Overall, these results are well-predicted by our theory. Moreover, it is remarkable to notice that this constant becomes large once increasing the depth. This mainly contributes to the fact that deeper neural network has higher non-linearity.

**Effects of width:** In this experiment, we explore the effect of width for 2-layer ReLU network with SGD optimizer (the initialization and the stepsize are the same as above). For Figure 4d, 4e, 4e, we set $a = 1.3, 1.0, 1.0$, respectively. As can be seen, the wider the neural network, the better our risk decomposition works. We comment that this behavior also comes to the level of non-linearity. It is now well-known that when the width tends to infinity, the neural network converges to a linear function (e.g. (Jacot et al., 2018)).

**Effects of optimizer:** In this experiment, we try three different optimizers: SGD (the same hyperparameters as before), Adam (stepsize $\eta = 0.002$, $(\beta_1, \beta_2) = (0.9, 0.999)$, $\epsilon = 1e - 08$, no weight decay), Rprop (learning rate $\eta = 5 \times 10^{-4}$, $(\eta_1, \eta_2) = (0.5, 1.2)$, stepsizes $(1 \times 10^{-6}, 50)$). For Figure 4g, 4h, 4i, we set $a = 1.3, 1.2, 1.1$. It can be seen that our risk decomposition framework is robust to different optimization algorithms, demonstrating its generality and potential to extend to general optimization problems.

## A.2   CIFAR-10

In this section we deliver additional experiment in CIFAR-10 dataset. In this experiment, we use ResNet-18 and choose Adam as the optimizer. We regard that optimizing over the original dataset as the bias training (in the sense that all the label is noiseless, referring to the bias part). As for the variance training, we set the ground truth to be the uniform logit $\mathbf{v} = [0.1, \cdots, 0.1]^\top$ (here we use one-hot coding and choose the categorical cross entropy loss). As for the labels in the training dataset, we generate it uniformly from $\{\mathbf{e}_1, \cdots, \mathbf{e}_{10}\}$, where $\mathbf{e}_i$ is the standard basis in $\mathbb{R}^{10}$. Under such data generating model, its Bayes risk is provided by $-10 \times 0.1 \times \log(0.1) \approx 2.3026$ for categorical cross entropy loss. For the standard training, we generate the training label as follows: $\mathbf{y} = (1-p)\mathbf{y}^\star + p\mathbf{e}_j$. Here $p \in [0, 1]$ is the corruption probability, $\mathbf{y}^\star$ is the true label and $j$ is uniformly sampled from $\{1, \cdots, 10\}$. In this experiment, we set the corruption probability to be $0.4$. The numerical result is provided in Figure 5 . Different from the performance on MINIST, here the variance training actually converges to the ground truth. We conjecture that this is due to the structure of ResNet, resulting to some kind of implicit regularization towards the training process.[10]

## B   Figure Illustration

### B.1   Figure 1a

Figure 1a conveys an important message that variance training has a much smaller gradient norm at the beginning compared to standard training. This mainly comes from the difference in their landscapes. Since there is no signal term in the variance learning, we can regard the landscape corresponding to the variance training as just a translation transformation of the original landscape (Figure 1a, the blue line is just a translation transformation of the green line). Based on this difference, they pursue different optima. We denote $\hat{\boldsymbol{\theta}}_v, \hat{\boldsymbol{\theta}}$ the empirical loss minimizer, respectively. Due to concentration, we have $\hat{\boldsymbol{\theta}}_v \approx 0$, and $\hat{\boldsymbol{\theta}} \approx \boldsymbol{\theta}^\star$. Note that we use a near-zero initialization $\boldsymbol{\theta}^{(0)} = \boldsymbol{\theta}_v^{(0)} \approx 0$, we immediately have that $\boldsymbol{\theta}_v^{(0)} \approx \hat{\boldsymbol{\theta}}_v$, indicating that the gradient norm is small. On the other hand, since $\boldsymbol{\theta}^{(0)}$ is far away from the optimum, its initial gradient norm can be very large.

To better illustrate this principle, we use linear regression to provide a simple example. For the standard training, the empirical loss is $\mathcal{L}_n(\boldsymbol{\theta}; \mathcal{P}) = \frac{1}{n} \sum_{i=1}^n (\langle \boldsymbol{\theta}, X_i \rangle - \langle \boldsymbol{\theta}^\star, X_i \rangle - \varepsilon_i)^2$. Hence, we

---

[10]We do observe that for certain learning rate regimes, the algorithm will overfit the training dataset for variance training, leading to a large VER ($\sim 10$).

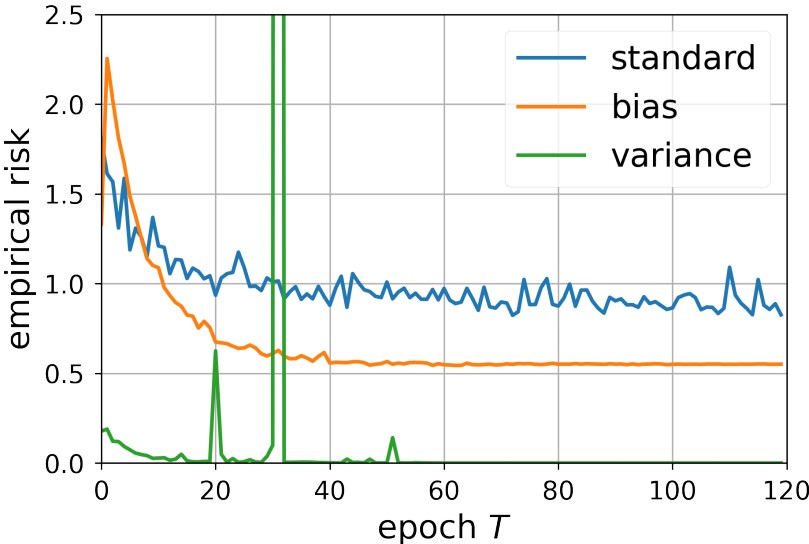

Figure 5: Three excess risk dynamics for CIFAR-10.

have

$$\nabla \mathcal{L}_n(\boldsymbol{\theta}^{(0)}; \mathcal{P}) = \frac{2}{n} \sum_{i=1}^{n} \left( \left\langle \boldsymbol{\theta}^{(0)}, X_i \right\rangle - \left\langle \boldsymbol{\theta}^\star, X_i \right\rangle - \varepsilon_i \right) X_i$$
$$= -\frac{2}{n} \sum_{i=1}^{n} \left( \left\langle \boldsymbol{\theta}^\star, X_i \right\rangle + \varepsilon_i \right) X_i \approx -2\boldsymbol{\theta}^\star, \tag{10}$$

if we assume $X_i \sim \mathcal{N}\left(0, I_{d \times d}\right)$. On the other hand, for the variance training, the empirical loss is $\mathcal{L}_n(\boldsymbol{\theta}; \mathcal{P}_v) = \frac{1}{n} \sum_{i=1}^{n} \left( \left\langle \boldsymbol{\theta}, X_i \right\rangle - \varepsilon_i \right)^2$. Therefore, we have

$$\nabla \mathcal{L}_n(\boldsymbol{\theta}_b^{(0)}; \mathcal{P}_v) = \frac{2}{n} \sum_{i=1}^{n} \left( \left\langle \boldsymbol{\theta}_b^{(0)}, X_i \right\rangle - \varepsilon_i \right) X_i = -\frac{2}{n} \sum_{i=1}^{n} \varepsilon_i X_i \approx 0. \tag{11}$$

In conclusion, we have

$$\left\| \nabla \mathcal{L}_n(\boldsymbol{\theta}^{(0)}; \mathcal{P}) \right\| \approx 2 \left\| \boldsymbol{\theta}^\star \right\|, \quad \left\| \nabla \mathcal{L}_n(\boldsymbol{\theta}_v^{(0)}; \mathcal{P}_v) \right\| \approx 0. \tag{12}$$

Obviously, there is a huge initial gradient norm gap between standard and variance training, showing that our bound is significantly better than the direct stability-based bound under such regimes.

## B.2 FIGURE 1B

Figure 1b provides an example showing that the trajectory experienced by the variance training is likely to stay a convex region, though the global landscape can be highly non-smooth and non-convex. The reason is similar to that of Figure 1a. Since $\boldsymbol{\theta}_v^{(0)} \approx \hat{\boldsymbol{\theta}}_v$, the gradient norm is small and the training dynamics may stay close to the optimum, which lies on a smooth and convex region. On the contrary, the standard training needs to cross potentially many bad local minimum or saddle points, suffering from large cumulative gradient norms.

## B.3 FIGURE 2

The setting of Figure 2 is similar to that of Section A. For MINIST dataset, we use three layer convoluted neural network with Adam optimizer.

## B.4 FIGURE 3

Figure 3 provides the numerical verification of Dynamics Decomposition Condition, and our upper bounds for both linear regression and matrix recovery problems [11].

**Linear Regression**    We set the dimension $d = 500$, the sample sizes are $n = 300, 800$, respectively. Samples $x_1 \cdots, x_n$ are i.i.d. from standard Gaussian distribution $\mathcal{N}(0, I_{d \times d})$, and the output $y_i = \langle x_i, \theta^\star \rangle + \varepsilon_i$ where $\varepsilon_i \sim \mathcal{N}(0, \sigma^2)$ and $\sigma = 2$. We set the initialization $\theta^{(0)} = \theta_b^{(0)} = \theta_v^{(0)} = 0$ and the learning rate $\lambda = 0.01$.

**Matrix Recovery**    We set the dimension $d = 20$, rank $r = 3$, the ground truth $X^\star = V \Sigma V^\top$, where $\Sigma = \text{Diag}\{5, 3, 1, 0, \cdots, 0\}$ and $V$ is an orthonormal matrix. The numbers of the measurement matrices are $200, 600$, respectively. The measurement matrices $A_1, \cdots, A_n$ are i.i.d. standard Gaussian matrices. The corresponding measurement $y_i = \langle A_i, X^\star \rangle + \varepsilon_i$, where $\varepsilon_i \sim \mathcal{N}(0, 1)$. In this numerical experiment, we use the initialization $U(0) = U_b(0) = U_v(0) = \alpha I_{d \times d}$ where $\alpha = 0.01$. We solve this problem via gradient descent with constant stepsize $\eta = 0.1$.

For the upper bound, it is calculated via upper bound $= \left\| X_t^b - X^\star \right\|_F + \left\| X_t^v \right\|_F + \frac{8.5}{\sqrt{n}}$, where the last term is corresponding to the additional term in Lemma 4.

**Neural Network**    We the dimension $d = 30$, the ground truth function is $f^\star(x) = \langle \theta^\star, x \rangle$ where $\theta^\star$ is sparse and $\|\theta^\star\| = \Theta(1)$. The covariate $x$ is generated from Gaussian distribution. The label for bias training is exactly $y_i = f^\star(x_i)$. For the variance trainng, its label is generated from an additional Gaussian distribution $\varepsilon_i \sim \mathcal{N}(0, \sigma^2)$ where $\sigma = 1.5$. For the standard training, its label is $y_i = f^\star(x_i) + \varepsilon_i$. For the upper bound, it is calculated via upper bound $= \left\| f_{\theta_t^b} - f^\star \right\|_\mathcal{P} + \left\| f_{\theta_t^v} \right\|_\mathcal{P}$. Here $\|\cdot\|_\mathcal{P}$ is defined to be the $L_2(\mathcal{P})$ norm, i.e., $\|g\|_\mathcal{P}^2 = \mathbb{E}_{x \sim \mathcal{P}}\left[ f(x)^2 \right]$. For these settings, we choose SGD (with stepsize $\eta = 1 \times 10^{-3}$) as the optimizer where the initialization is randomly generated from a Gaussian distribution.

## C    PROOF FOR OVERPARAMETERIZED LINEAR REGRESSION

### C.1    PROOF OF THEOREM 1

In this section, we prove Theorem 1 stated as follows:

**Theorem 1 (Linear Regression Regimes)** *Under the overparameterized linear regression settings, assume that $w = \frac{\theta^{*,\top} x}{\sqrt{\theta^{*,\top} \Sigma_x \theta^*}}$ is $\sigma_w^2$-subgaussian. If $\|x\| \leq 1$, $|\epsilon| \leq V$, $\sup_{t \in [T]} \|\hat{\theta}_v^{(t)}\| \leq B$ are all bounded, the following inequality holds with probability at least $1 - \delta$:*

$$\mathcal{E}_\mathcal{L}(\hat{\theta}^{(T)}; \mathcal{P}) = \tilde{\mathcal{O}} \left( \max\left\{ 1, \theta^{*,\top} \Sigma_x \theta^* \sigma_w^2, [V + B]^2 \right\} \sqrt{\frac{\log(4/\delta)}{n}} + \frac{\|\theta^*\|^2}{\lambda T} + \frac{T\lambda[V + B]^2}{n} \right),$$

*where the probability is taken over the randomness of training data.*

**Proof:**

---

[11]Here, we extend the diagonal matrix recovery problem to the general matrix recovery problem.

We first calculate the population loss under linear regression regime, namely, for any time $t$:

$$\mathcal{L}(\hat{\boldsymbol{\theta}}^{(t)}; \mathcal{P}) = \mathbb{E}_{\boldsymbol{x}, \boldsymbol{y}} \left[ \boldsymbol{y} - \boldsymbol{x}^\top \hat{\boldsymbol{\theta}}^{(t)} \right]^2$$

$$\overset{(a)}{=} \mathbb{E} \left[ \boldsymbol{x}^\top \boldsymbol{\theta}^* + \epsilon - \boldsymbol{x}^\top \hat{\boldsymbol{\theta}}^{(t)} \right]^2$$

$$= \mathbb{E} \left[ \boldsymbol{x}^\top \boldsymbol{\theta}^* - \boldsymbol{x}^\top \hat{\boldsymbol{\theta}}^{(t)} \right]^2 - 2\mathbb{E} \left[ \epsilon \boldsymbol{x}^\top \hat{\boldsymbol{\theta}}^{(t)} \right] + \mathbb{E}\epsilon^2$$

$$= \left[ \boldsymbol{\theta}^* - \hat{\boldsymbol{\theta}}^{(t)} \right]^\top \Sigma_x \left[ \boldsymbol{\theta}^* - \hat{\boldsymbol{\theta}}^{(t)} \right] - 2\mathbb{E}_{\boldsymbol{x}} \mathbb{E}_\epsilon \left[ \epsilon \boldsymbol{x}^\top \hat{\boldsymbol{\theta}}^{(t)} | \boldsymbol{x} \right] + \mathbb{E}\epsilon^2$$

$$= \left[ \boldsymbol{\theta}^* - \hat{\boldsymbol{\theta}}^{(t)} \right]^\top \Sigma_x \left[ \boldsymbol{\theta}^* - \hat{\boldsymbol{\theta}}^{(t)} \right] - 2\mathbb{E}_{\boldsymbol{x}} \left[ \boldsymbol{x}^\top \hat{\boldsymbol{\theta}}^{(t)} \mathbb{E}_\epsilon \left[ \epsilon | \boldsymbol{x} \right] \right] + \mathbb{E}\epsilon^2$$

$$\overset{(b)}{=} \left[ \boldsymbol{\theta}^* - \hat{\boldsymbol{\theta}}^{(t)} \right]^\top \Sigma_x \left[ \boldsymbol{\theta}^* - \hat{\boldsymbol{\theta}}^{(t)} \right] + \mathbb{E}\epsilon^2,$$

where (a) we use the condition that $\boldsymbol{y} = \boldsymbol{x}^\top \boldsymbol{\theta}^* + \epsilon$ and (b) we use the assumption that $\mathbb{E}[\epsilon|\boldsymbol{x}] = 0$.

Therefore, the excess risk can be written as

$$\mathcal{E}(\hat{\boldsymbol{\theta}}^{(t)}) = \left[ \boldsymbol{\theta}^* - \hat{\boldsymbol{\theta}}^{(t)} \right]^\top \Sigma_x \left[ \boldsymbol{\theta}^* - \hat{\boldsymbol{\theta}}^{(t)} \right].$$

Now plugging the Lemma 1, Lemma 2, Lemma 3, into Theorem 2, we derive that with probability at least $1 - \delta$:

$$\mathcal{E}_{\mathcal{L}}(\hat{\boldsymbol{\theta}}^{(T)}; \mathcal{P}) \lesssim \mathcal{E}_{\mathcal{L}}^v(\hat{\boldsymbol{\theta}}_v^{(T)}; \mathcal{P}) + \mathcal{E}_{\mathcal{L}}^b(\hat{\boldsymbol{\theta}}_b^{(T)}; \mathcal{P})$$

$$\lesssim \frac{\boldsymbol{\theta}^{*,\top} \Sigma_{\boldsymbol{x}} \boldsymbol{\theta}^*}{\sqrt{n}} \sigma_w^2 \sqrt{\log\left(\frac{4}{\delta}\right)} + \frac{1}{\lambda[2T+1]} \|\boldsymbol{\theta}^*\|^2$$

$$+ \frac{T\lambda}{n} [V+B]^2 \log(n) \log(2n/\delta) + \max\left\{1, [V+B]^2\right\} \sqrt{\frac{\log(4/\delta)}{2n}}.$$

$$= \tilde{\mathcal{O}} \left( \max\left\{1, \boldsymbol{\theta}^{*,\top} \Sigma_{\boldsymbol{x}} \boldsymbol{\theta}^* \sigma_w^2, [V+B]^2\right\} \sqrt{\frac{\log(4/\delta)}{n}} + \frac{\|\boldsymbol{\theta}^*\|^2}{\lambda T} + \frac{T\lambda[V+B]^2}{n} \right),$$

which completes the proof. $\qquad\qquad\qquad\qquad\qquad\qquad\qquad\qquad\qquad\qquad\qquad\qquad\square$

## C.2    PROOF OF LEMMA 1

We first recall Lemma 1 as follows:

**Lemma 1** *Under the overparameterized linear regression settings with initialization $\boldsymbol{\theta}^{(0)} = \boldsymbol{\theta}_b^{(0)} = \boldsymbol{\theta}_v^{(0)} = \mathbf{0}$, for any time T, the following decomposition inequality holds:*

$$\mathcal{E}_{\mathcal{L}}(\hat{\boldsymbol{\theta}}^{(T)}; \mathcal{P}) \leq 2\mathcal{E}_{\mathcal{L}}^v(\hat{\boldsymbol{\theta}}_v^{(T)}; \mathcal{P}) + 2\mathcal{E}_{\mathcal{L}}^b(\hat{\boldsymbol{\theta}}_b^{(T)}; \mathcal{P}).$$

**Proof:** Firstly, when we use gradient descent, we have that:

$$\hat{\boldsymbol{\theta}}^{(t+1)} = \hat{\boldsymbol{\theta}}^{(t)} + \frac{\lambda}{n} \sum_i \boldsymbol{x}_i [\boldsymbol{y}_i - \boldsymbol{x}_i^\top \hat{\boldsymbol{\theta}}^{(t)}],$$

$$\hat{\boldsymbol{\theta}}_v^{(t+1)} = \hat{\boldsymbol{\theta}}_v^{(t)} + \frac{\lambda}{n} \sum_i \boldsymbol{x}_i [\epsilon_i - \boldsymbol{x}_i^\top \hat{\boldsymbol{\theta}}_v^{(t)})],$$

$$\hat{\boldsymbol{\theta}}_b^{(t+1)} = \hat{\boldsymbol{\theta}}_b^{(t)} + \frac{\lambda}{n} \sum_i \boldsymbol{x}_i [\boldsymbol{x}_i^\top \boldsymbol{\theta}^* - \boldsymbol{x}_i^\top \hat{\boldsymbol{\theta}}_b^{(t)}].$$

We next prove the conclusion by induction. Since we choose $\boldsymbol{\theta}^{(0)} = \boldsymbol{\theta}_v^{(0)} = \boldsymbol{\theta}_b^{(0)} = 0$, the conclusion holds at the initialization

$$\boldsymbol{\theta}^{(0)} = \boldsymbol{\theta}_v^{(0)} + \boldsymbol{\theta}_b^{(0)}.$$

Now assume that $\hat{\boldsymbol{\theta}}^{(t)} = \hat{\boldsymbol{\theta}}_v^{(t)} + \hat{\boldsymbol{\theta}}_b^{(t)}$ holds at time $t$, we have

$$
\begin{aligned}
\hat{\boldsymbol{\theta}}^{(t+1)} &= \hat{\boldsymbol{\theta}}^{(t)} + \frac{\lambda}{n} \sum_i \boldsymbol{x}_i [\boldsymbol{y}_i - \boldsymbol{x}_i^\top \hat{\boldsymbol{\theta}}^{(t)}] \\
&= \hat{\boldsymbol{\theta}}_v^{(t)} + \hat{\boldsymbol{\theta}}_b^{(t)} + \frac{\lambda}{n} \sum_i \boldsymbol{x}_i \left[ \boldsymbol{\epsilon}_i + \boldsymbol{x}_i^\top \boldsymbol{\theta}^* - \boldsymbol{x}_i^\top \left( \hat{\boldsymbol{\theta}}_v^{(t)} + \hat{\boldsymbol{\theta}}_b^{(t)} \right) \right] \\
&= \left[ \hat{\boldsymbol{\theta}}_v^{(t)} + \frac{\lambda}{n} \sum_i \boldsymbol{x}_i [\boldsymbol{\epsilon}_i - \boldsymbol{x}_i^\top \hat{\boldsymbol{\theta}}_v^{(t)}] \right] + \left[ \hat{\boldsymbol{\theta}}_b^{(t)} + \frac{\lambda}{n} \sum_i \boldsymbol{x}_i [\boldsymbol{x}_i^\top \boldsymbol{\theta}^* - \boldsymbol{x}_i^\top \hat{\boldsymbol{\theta}}_b^{(t)}] \right] \\
&= \hat{\boldsymbol{\theta}}_v^{(t+1)} + \hat{\boldsymbol{\theta}}_b^{(t+1)}.
\end{aligned}
$$

In summary, we have: $\hat{\boldsymbol{\theta}}^{(t)} = \hat{\boldsymbol{\theta}}_v^{(t)} + \hat{\boldsymbol{\theta}}_b^{(t)}$ holds for any time $t$. Besides, we easily calculate that $\boldsymbol{\theta}^* = \boldsymbol{\theta}_v^* + \boldsymbol{\theta}_b^*$ since $\boldsymbol{\theta}^* = \boldsymbol{\theta}_b^* = \boldsymbol{\theta}^*$ and $\boldsymbol{\theta}_v^* = \boldsymbol{0}$.

Therefore, we have $\hat{\boldsymbol{\theta}}^{(t)} - \boldsymbol{\theta}^\star = \hat{\boldsymbol{\theta}}_v^{(t)} - \boldsymbol{\theta}_v^\star + \hat{\boldsymbol{\theta}}_b^{(t)} - \boldsymbol{\theta}_b^\star$ holds for any $t$. By Cauchy–Schwarz inequality, we derive that for a specific time $T$:

$$
\begin{aligned}
\mathcal{E}_{\mathcal{L}}&(\hat{\boldsymbol{\theta}}^{(T)}; \mathcal{P}) \\
&= \left[ \boldsymbol{\theta}^* - \hat{\boldsymbol{\theta}}^{(T)} \right]^\top \Sigma_x \left[ \boldsymbol{\theta}^* - \hat{\boldsymbol{\theta}}^{(T)} \right] \\
&= \left\| \boldsymbol{\theta}^* - \hat{\boldsymbol{\theta}}^{(T)} \right\|_{\Sigma_x}^2 \\
&= \left\| \boldsymbol{\theta}_v^* - \hat{\boldsymbol{\theta}}_v^{(T)} + \boldsymbol{\theta}_b^* - \hat{\boldsymbol{\theta}}_b^{(T)} \right\|_{\Sigma_x}^2 \\
&\overset{(a)}{\leq} 2 \left\| \boldsymbol{\theta}_v^* - \hat{\boldsymbol{\theta}}_v^{(T)} \right\|_{\Sigma_x}^2 + 2 \left\| \boldsymbol{\theta}_b^* - \hat{\boldsymbol{\theta}}_b^{(T)} \right\|_{\Sigma_x}^2 \\
&\leq 2 \mathcal{E}_{\mathcal{L}}^v(\hat{\boldsymbol{\theta}}_v^{(T)}; \mathcal{P}) + 2 \mathcal{E}_{\mathcal{L}}^b(\hat{\boldsymbol{\theta}}_b^{(T)}; \mathcal{P}).
\end{aligned}
$$

where we denote the norm $\|u\|_A^2 = u^\top A u$.

$\square$

## C.3  PROOF OF LEMMA 2

**Lemma 2** *Under the overparameterized linear regression settings, assume that $\|\boldsymbol{x}\| \leq 1$, $|\epsilon| \leq V$, $\sup_t \|\hat{\boldsymbol{\theta}}_v^{(t)}\| \leq B$ are all bounded. Consider the gradient descent algorithm with constant stepsize $\lambda$, with probability at least $1 - \delta$:*

$$
\mathcal{E}_{\mathcal{L}}^v \left( \hat{\boldsymbol{\theta}}_v^{(T)}; \mathcal{P} \right) = \mathcal{O} \left( \frac{T\lambda}{n} [V + B]^2 \log(n) \log(2n/\delta) + \max\left\{ 1, [V + B]^2 \right\} \sqrt{\frac{\log(2/\delta)}{2n}} \right). \quad (2)
$$

**Proof:** When the context is clear, we omit $v$ in this section and denote the trained parameter by $\hat{\boldsymbol{\theta}}^{(t)}$. For any time $t$, we first split the excess risk into three components following Equation 1.

$$
\mathcal{E}_{\mathcal{L}}(\hat{\boldsymbol{\theta}}^{(t)}; \mathcal{P}_v) = \underbrace{\left[ \mathcal{L}(\hat{\boldsymbol{\theta}}^{(t)}; \mathcal{P}_v) - \mathcal{L}(\hat{\boldsymbol{\theta}}^{(t)}; \mathcal{P}_v^n) \right]}_{(I)} + \underbrace{\left[ \mathcal{L}(\hat{\boldsymbol{\theta}}^{(t)}; \mathcal{P}_v^n) - \mathcal{L}(\boldsymbol{\theta}^*; \mathcal{P}_v^n) \right]}_{(II)} + \underbrace{\left[ \mathcal{L}(\boldsymbol{\theta}^*; \mathcal{P}_v^n) - \mathcal{L}(\boldsymbol{\theta}^*; \mathcal{P}_v) \right]}_{(III)}.
$$

$$(13)$$

**Part (I).** We bound the first component in Equation 13 using stability.

**Fact A:** the loss $\ell(\boldsymbol{\theta}; \boldsymbol{x}, \boldsymbol{y})$ is $2[V + B]$-Lipschitz.

$$
\|\nabla_{\boldsymbol{\theta}} \ell(\boldsymbol{\theta}; \boldsymbol{x}, \boldsymbol{y})\| = 2\|\boldsymbol{x}(\boldsymbol{\epsilon} - \boldsymbol{x}^\top \boldsymbol{\theta})\| \leq 2[V + B].
$$

**Fact B:** the stability of the parameter is $\frac{2\lambda t}{n}[V + B]$. Denote the parameter trained on dataset $S$ and the parameter trained on dataset $S'$ as $\hat{\boldsymbol{\theta}}_S^{(t)}$ and $\hat{\boldsymbol{\theta}}_{S'}^{(t)}$, respectively. We will show that, when $S$ and $S'$

have only one different sample (e.g., WLOG $S = \{x_1, x_2, \cdots, x_n\}$ and $S' = \{x'_1, x_2, \cdots, x_n\}$), $\hat{\boldsymbol{\theta}}_S^{(t)}$ is close to $\hat{\boldsymbol{\theta}}_{S'}^{(t)}$.

$$
\left\| \hat{\boldsymbol{\theta}}_S^{(t+1)} - \hat{\boldsymbol{\theta}}_{S'}^{(t+1)} \right\|
$$

$$
\leq \left\| \left[ I - \frac{\lambda}{n} X^\top X \right] \hat{\boldsymbol{\theta}}_S^{(t)} + \frac{\lambda}{n} X^\top Y - \left[ I - \frac{\lambda}{n} X'^{,\top} X' \right] \hat{\boldsymbol{\theta}}_{S'}^{(t)} - \frac{\lambda}{n} X'^{,\top} Y' \right\|
$$

$$
\leq \left\| \left[ I - \frac{\lambda}{n} X^\top X \right] \left[ \hat{\boldsymbol{\theta}}_S^{(t)} - \hat{\boldsymbol{\theta}}_{S'}^{(t)} \right] \right\| + \left\| \left[ \frac{\lambda}{n} X'^{,\top} X' - \frac{\lambda}{n} X^\top X \right] \hat{\boldsymbol{\theta}}_{S'}^{(t)} \right\| + \frac{\lambda}{n} \left\| X^\top Y - X'^{,\top} Y' \right\|
$$

$$
\leq \left\| \left[ \hat{\boldsymbol{\theta}}_S^{(t)} - \hat{\boldsymbol{\theta}}_{S'}^{(t)} \right] \right\| + \frac{\lambda}{n} \left\| \left[ x_1 x_1^\top - x'_1 x_1^{\top,'} \right] \hat{\boldsymbol{\theta}}_{S'}^{(t)} \right\| + \frac{\lambda}{n} \left\| x_1^\top \epsilon_1 - x_1'^{,\top} \epsilon'_1 \right\|
$$

$$
\leq \left\| \left[ \hat{\boldsymbol{\theta}}_S^{(t)} - \hat{\boldsymbol{\theta}}_{S'}^{(t)} \right] \right\| + \frac{2\lambda}{n} \left[ V + B \right].
$$

Summation over time $t$ provides the following result:

$$
\| \hat{\boldsymbol{\theta}}_S^{(t)} - \hat{\boldsymbol{\theta}}_{S'}^{(t)} \| \leq \frac{2t\lambda}{n} \left[ V + B \right].
$$

Therefore, the total stability with respect to the $\ell_2$-loss is

$$
|\ell(\hat{\boldsymbol{\theta}}_S^{(t)}) - \ell(\hat{\boldsymbol{\theta}}_{S'}^{(t)})| \leq \frac{4t\lambda}{n} \left[ V + B \right]^2.
$$

By Proposition 1, we have that

$$
\mathcal{L}(\hat{\boldsymbol{\theta}}^{(t)}; \mathcal{P}_v) - \mathcal{L}(\hat{\boldsymbol{\theta}}^{(t)}; \mathcal{P}_v^n) \lesssim \frac{t\lambda}{n} \left[ V + B \right]^2 \log(n) \log(2n/\delta) + \sqrt{\frac{\log(1/\delta)}{n}}.
$$

**Part II.** We next show that the second component in Equation 13 is less than zero based on Lemma 7.

$$
\mathcal{L}(\boldsymbol{\theta}^{(t)}; \mathcal{P}_v^n) - \mathcal{L}(\boldsymbol{\theta}^*; \mathcal{P}_v^n)
$$

$$
= \frac{1}{n} \left[ \| \epsilon - X\boldsymbol{\theta}^{(t)} \|^2 - \| \epsilon \|^2 \right]
$$

$$
= \frac{1}{n} \epsilon^\top \left[ -2X \left[ I - \left[ I - \frac{\lambda}{n} X^\top X \right]^t \right] X^\dagger + X^{\dagger,\top} \left[ I - \left[ I - \frac{\lambda}{n} X^\top X \right]^t \right] X^\top X \left[ I - \left[ I - \frac{\lambda}{n} X^\top X \right]^t \right] X^\dagger \right] \epsilon.
$$

The $i$-th eigenvalue of the matrix $[-2X[I - [I - \frac{\lambda}{n}X^\top X]^t]X^\dagger + X^{\dagger,\top}[I - [I - \frac{\lambda}{n}X^\top X]^t]X^\top X[I - [I - \frac{\lambda}{n}X^\top X]^t]X^\dagger]$ can be calculated as follows

$$
-2 \left[ 1 - (1 - \lambda\sigma_i)^t \right] + \left[ 1 - (1 - \lambda\sigma_i)^t \right]^2 = (1 - \lambda\sigma_i)^{2t} - 1 \leq 0,
$$

where $\sigma_i$ refers to the $i$-th eigenvalue of the matrix $\frac{1}{n}X^\top X$. Therefore, we have

$$
\mathcal{L}(\boldsymbol{\theta}^{(t)}; \mathcal{P}_v^n) - \mathcal{L}(\boldsymbol{\theta}^*; \mathcal{P}_v^n) \leq 0.
$$

**Part III.** We bound the third component in Equation 13 using concentration.

Note that $|\ell_2(\boldsymbol{\theta}; x, y)| = |(y - x^\top \boldsymbol{\theta})^2| \leq [V + B]^2$, so we can use Hoeffding's inequality,

$$
\mathbb{P}\left[ |\mathcal{L}(\boldsymbol{\theta}^*; \mathcal{P}_v^n) - \mathcal{L}(\boldsymbol{\theta}^*; \mathcal{P}_v)| \geq u \right] \leq 2 \exp\left[ -\frac{2nu^2}{(B + V)^4} \right].
$$

By setting $u = [V + B]^2 \sqrt{\frac{\log(2/\delta)}{2n}}$, then with probability at least $1 - \delta$, we have

$$
|\mathcal{L}(\boldsymbol{\theta}^*; \mathcal{P}_v^n) - \mathcal{L}(\boldsymbol{\theta}^*; \mathcal{P}_v)| \leq [V + B]^2 \sqrt{\frac{\log(2/\delta)}{2n}}.
$$

Combining Part (I), (II), and (III) together, at time $T$, we have:

$$\mathcal{E}_{\mathcal{L}}(\hat{\boldsymbol{\theta}}^{(T)}; \mathcal{P}_v) \lesssim \frac{T\lambda}{n}[V+B]^2 \log(n)\log(2n/\delta) + \sqrt{\frac{\log(1/\delta)}{n}} + [V+B]^2\sqrt{\frac{\log(2/\delta)}{2n}}$$

$$\lesssim \frac{T\lambda}{n}[V+B]^2 \log(n)\log(2n/\delta) + \max\{1, [V+B]^2\}\sqrt{\frac{\log(2/\delta)}{2n}}.$$

$\square$

### C.4 PROOF OF LEMMA 3

**Lemma 3** *Under the overparameterized linear regression settings, assume that* $w = \frac{\boldsymbol{\theta}^{*,\top}\boldsymbol{x}}{\sqrt{\boldsymbol{\theta}^{*,\top}\Sigma_{\boldsymbol{x}}\boldsymbol{\theta}^{*}}}$ *is* $\sigma_w^2$*-subgaussian, then with probability at least* $1 - \delta$*:*

$$\mathcal{E}_{\mathcal{L}}^{b}(\hat{\boldsymbol{\theta}}_{b}^{(T)}; \mathcal{P}) = \mathcal{O}\left(\frac{\boldsymbol{\theta}^{*,\top}\Sigma_{\boldsymbol{x}}\boldsymbol{\theta}^{*}}{\sqrt{n}}\sigma_w^2\sqrt{\log\left(\frac{2}{\delta}\right)} + \frac{1}{\lambda[2T+1]}\|\boldsymbol{\theta}^{*}\|^2\right). \tag{3}$$

**Proof:** Since we use the noiseless data $(\boldsymbol{x}, \mathbb{E}(\boldsymbol{y}|\boldsymbol{x}))$ and $\mathbb{E}(\boldsymbol{y}|\boldsymbol{x}) = \boldsymbol{x}^\top\boldsymbol{\theta}^*$, by Lemma 7 we have

$$\boldsymbol{\theta}_{b}^{(t)} = \left[I - \left[I - \frac{\lambda}{n}X^\top X\right]^t\right]X^\dagger X\boldsymbol{\theta}^* = \boldsymbol{\theta}^*\left[I - \left[I - \frac{\lambda}{n}X^\top X\right]^t\right]\boldsymbol{\theta}^*,$$

where we use the fact that $\left[I - \left[I - \frac{\lambda}{n}X^\top X\right]^t\right]X^\dagger X = I - \left[I - \frac{\lambda}{n}X^\top X\right]^t$.

Therefore, BER can be expressed as:

$$\mathcal{E}_{\mathcal{L}}(\boldsymbol{\theta}; \mathcal{P}_b^n) = \left[\boldsymbol{\theta}^* - \boldsymbol{\theta}_b^{(t)}\right]^\top\Sigma_{\boldsymbol{x}}\left[\boldsymbol{\theta}^* - \boldsymbol{\theta}_b^{(t)}\right]$$

$$= \boldsymbol{\theta}^{*,\top}\left[I - \frac{\lambda}{n}X^\top X\right]^t\Sigma_{\boldsymbol{x}}\left[I - \frac{\lambda}{n}X^\top X\right]^t\boldsymbol{\theta}^*$$

$$= \boldsymbol{\theta}^{*,\top}\left[I - \frac{\lambda}{n}X^\top X\right]^t\left[\Sigma_{\boldsymbol{x}} - \frac{1}{n}X^\top X\right]\left[I - \frac{\lambda}{n}X^\top X\right]^t\boldsymbol{\theta}^*$$

$$+ \boldsymbol{\theta}^{*,\top}\left[I - \frac{\lambda}{n}X^\top X\right]^t\left[\frac{1}{n}X^\top X\right]\left[I - \frac{\lambda}{n}X^\top X\right]^t\boldsymbol{\theta}^*.$$

We split the excess risk into two parts. Intuitively, it is just like the decomposition in Equation 1. The first part is similar to the generalization gap to use uniform convergence to bound it. Note that here we require that $\lambda < \sup_{X_0}\frac{1}{\|\frac{1}{n}X_0^\top X_0\|}$. Hence, we have

$$\left| \boldsymbol{\theta}^{*,\top} \left[ I - \frac{\lambda}{n} X^\top X \right]^t \left[ \Sigma_{\boldsymbol{x}} - \frac{1}{n} X^\top X \right] \left[ I - \frac{\lambda}{n} X^\top X \right]^t \boldsymbol{\theta}^* \right|$$

$$\leq \left| \sup_{X_0} \boldsymbol{\theta}^{*,\top} \left[ I - \frac{\lambda}{n} X_0{}^\top X_0 \right]^t \left[ \Sigma_{\boldsymbol{x}} - \frac{1}{n} X^\top X \right] \left[ I - \frac{\lambda}{n} X_0{}^\top X_0 \right]^t \boldsymbol{\theta}^* \right|$$

$$= \left| \sup_{X_0} \operatorname{tr} \left( \boldsymbol{\theta}^{*,\top} \left[ I - \frac{\lambda}{n} X_0{}^\top X_0 \right]^t \left[ \Sigma_{\boldsymbol{x}} - \frac{1}{n} X^\top X \right] \left[ I - \frac{\lambda}{n} X_0{}^\top X_0 \right]^t \boldsymbol{\theta}^* \right) \right|$$

$$= \left| \sup_{X_0} \operatorname{tr} \left( \left[ I - \frac{\lambda}{n} X_0{}^\top X_0 \right]^t \left[ \Sigma_{\boldsymbol{x}} - \frac{1}{n} X^\top X \right] \left[ I - \frac{\lambda}{n} X_0{}^\top X_0 \right]^t \boldsymbol{\theta}^* \boldsymbol{\theta}^{*,\top} \right) \right|$$

$$= \left| \sup_{X_0} \left\| \left[ I - \frac{\lambda}{n} X_0{}^\top X_0 \right]^t \left[ \Sigma_{\boldsymbol{x}} - \frac{1}{n} X^\top X \right] \left[ I - \frac{\lambda}{n} X_0{}^\top X_0 \right]^t \boldsymbol{\theta}^* \boldsymbol{\theta}^{*,\top} \right\| \right|$$

$$\leq \left| \sup_{X_0} \left\| \left[ I - \frac{\lambda}{n} X_0{}^\top X_0 \right]^t \right\| \left\| \left[ \Sigma_{\boldsymbol{x}} - \frac{1}{n} X^\top X \right] \left[ I - \frac{\lambda}{n} X_0{}^\top X_0 \right]^t \boldsymbol{\theta}^* \boldsymbol{\theta}^{*,\top} \right\| \right|$$

$$\leq \left| \sup_{X_0} \left\| \left[ \Sigma_{\boldsymbol{x}} - \frac{1}{n} X^\top X \right] \left[ I - \frac{\lambda}{n} X_0{}^\top X_0 \right]^t \boldsymbol{\theta}^* \boldsymbol{\theta}^{*,\top} \right\| \right|$$

$$= \left| \sup_{X_0} \operatorname{tr} \left( \boldsymbol{\theta}^{*,\top} \left[ \Sigma_{\boldsymbol{x}} - \frac{1}{n} X^\top X \right] \left[ I - \frac{\lambda}{n} X_0{}^\top X_0 \right]^t \boldsymbol{\theta}^* \right) \right|$$

$$\leq \left| \sup_{X_0} \operatorname{tr} \left( \boldsymbol{\theta}^{*,\top} \left[ \Sigma_{\boldsymbol{x}} - \frac{1}{n} X^\top X \right] \boldsymbol{\theta}^* \right) \right|$$

$$= \left| \boldsymbol{\theta}^{*,\top} \left[ \Sigma_{\boldsymbol{x}} - \frac{1}{n} X^\top X \right] \boldsymbol{\theta}^* \right|,$$

where we apply Lemma 8 repeatedly since $\operatorname{rank} \left[ \boldsymbol{\theta}^{*,\top} \boldsymbol{\theta}^* A \right] = 1$ where $A$ stands for arbitrary matrix.

We highlight the difference between $X_0$ and $X$. The notation $X$ represents the contribution of the training set to the training error. The notation $X_0$ represents the contribution of the training set to the training estimator $\hat{\boldsymbol{\theta}}$. The uniform convergence is taken over the estimator $\boldsymbol{\theta}$. Therefore, we take sup operator on $X_0$.

We next bound $\boldsymbol{\theta}^{*,\top} \left[ \Sigma_{\boldsymbol{x}} - \frac{1}{n} X^\top X \right] \boldsymbol{\theta}^*$. Assume that $w = \boldsymbol{\theta}^{*,\top} \boldsymbol{x} / \sqrt{\boldsymbol{\theta}^{*,\top} \Sigma_{\boldsymbol{x}} \boldsymbol{\theta}^*}$ is SubGaussian with subGaussian norm $\sigma_w$. Obviously, $\mathbb{E}\left[w\right] = 0, \mathbb{E}\left[w^2\right] = 1$. Therefore, we have that: $w^2 - 1$ is sub-Exponential distributions with sub-Exponential norm $\left\| w^2 - 1 \right\|_{\psi_1} \leq \sigma_w^2$.

While $\epsilon / \left[ \boldsymbol{\theta}^\top \Sigma_{\boldsymbol{x}} \boldsymbol{\theta} \right] < \sigma_w^2$, we have the following tail bound

$$\mathbb{P}\left( |\boldsymbol{\theta}^{*,\top} \left[ \Sigma_{\boldsymbol{x}} - \frac{1}{n} X^\top X \right] \boldsymbol{\theta}^*| \geq \epsilon \right) = \mathbb{P}\left( |\frac{1}{n} \sum_{i=1}^n \left[ (\boldsymbol{x}_i^\top \boldsymbol{\theta})^2 - \boldsymbol{\theta}^\top \Sigma_{\boldsymbol{x}} \boldsymbol{\theta} \right] | \geq \epsilon \right)$$

$$= \mathbb{P}\left( |\frac{1}{n} \sum_{i=1}^n \left[ w_i^2 - 1 \right] | \geq \epsilon / \left[ \boldsymbol{\theta}^\top \Sigma_{\boldsymbol{x}} \boldsymbol{\theta} \right] \right)$$

$$\leq 2 \exp \left[ -c \frac{\left[ \epsilon / \left[ \boldsymbol{\theta}^\top \Sigma_{\boldsymbol{x}} \boldsymbol{\theta} \right] \right]^2}{n \sigma_w^4} \right],$$

where $c$ is a universal constant.

Therefore, by setting $\epsilon = \left[ \boldsymbol{\theta}^\top \Sigma_{\boldsymbol{x}} \boldsymbol{\theta} \right] \frac{\sigma_w^2 \log[2/\delta]}{\sqrt{c}\sqrt{n}}$ with probability at least $1 - \delta$, we have

$$\left| \boldsymbol{\theta}^{*,\top} \left[ \Sigma_{\boldsymbol{x}} - \frac{1}{n} X^\top X \right] \boldsymbol{\theta}^* \right| \leq \left[ \boldsymbol{\theta}^\top \Sigma_{\boldsymbol{x}} \boldsymbol{\theta} \right] \frac{\sigma_w^2 \sqrt{\log[2/\delta]}}{\sqrt{c}\sqrt{n}}.$$

With abuse to use $c$ as a constant, we have with probability at least $\geq 1 - \delta$

$$\left| \boldsymbol{\theta}^{*,\top} \left[ \Sigma_{\boldsymbol{x}} - \frac{1}{n} X^\top X \right] \boldsymbol{\theta}^* \right| \leq c \left[ \boldsymbol{\theta}^\top \Sigma_{\boldsymbol{x}} \boldsymbol{\theta} \right] \frac{\sigma_w^2 \sqrt{\log(2/\delta)}}{\sqrt{n}}.$$

For the second part, which is closely related to the empirical loss of $\boldsymbol{\theta}_b^{(t)}$, We will show that: for a general case, the empirical loss converges with rate $\mathcal{O}(1/t)$. Denote the eigenvalues of $\frac{1}{n} X^\top X$ as $\sigma_1, \cdots, \sigma_n, 0, \cdots, 0$. Then the corresponding eigenvalues of matrix $\left[ I - \frac{\lambda}{n} X^\top X \right]^t \left[ \frac{1}{n} X^\top X \right] \left[ I - \frac{\lambda}{n} X^\top X \right]^t$ are $(1 - \lambda \sigma_i)^{2t} \sigma_i$. Therefore, we have:

$$\boldsymbol{\theta}^{*,\top} \left[ I - \frac{\lambda}{n} X^\top X \right]^t \left[ \frac{1}{n} X^\top X \right] \left[ I - \frac{\lambda}{n} X^\top X \right]^t \boldsymbol{\theta}^*$$

$$\leq \|\boldsymbol{\theta}^*\|^2 \max_i (1 - \lambda \sigma_i)^{2t} \sigma_i$$

$$\leq \frac{1}{\lambda(2t+1)} \|\boldsymbol{\theta}^*\|^2.$$

The maximum is attained while $\sigma_i = \frac{1}{\lambda(2t+1)}$.

Combining these terms leads to the conclusion. $\qquad\qquad\square$

### C.5 AUXILIARY LEMMAS

**Lemma 7** *The dynamics of $\hat{\boldsymbol{\theta}}^{(t)}$ using GD is:*

$$\hat{\boldsymbol{\theta}}^{(t)} = \left[ I - \frac{\lambda}{n} X^\top X \right]^t \left[ \boldsymbol{\theta}^{(0)} - X^\dagger Y \right] + X^\dagger Y.$$

*where $X^\dagger$ represents the pseudo inverse of matrix $X$.*

**Proof:** The proof is based on induction. We notice that when $t = 0$, the equation directly follows. Assume when $t = t$, the equation holds. Then at time $t + 1$, we have

$$\hat{\boldsymbol{\theta}}^{(t+1)} = \hat{\boldsymbol{\theta}}^{(t)} + \frac{\lambda}{n} X^\top (Y - X \hat{\boldsymbol{\theta}}^{(t)})$$

$$= \left[ 1 - \frac{\lambda}{n} X^\top X \right] \left[ \left[ I - \frac{\lambda}{n} X^\top X \right]^t \left[ \boldsymbol{\theta}^{(0)} - X^\dagger Y \right] + X^\dagger Y \right] + \frac{\lambda}{n} X^\top Y$$

$$= \left[ I - \frac{\lambda}{n} X^\top X \right]^{t+1} \left[ \boldsymbol{\theta}^{(0)} - X^\dagger Y \right] + \left[ 1 - \frac{\lambda}{n} X^\top X \right] X^\dagger Y + \frac{\lambda}{n} X^\top Y$$

$$= \left[ I - \frac{\lambda}{n} X^\top X \right]^{t+1} \left[ \boldsymbol{\theta}^{(0)} - X^\dagger Y \right] + X^\dagger Y,$$

where we use the fact that $X^\top X X^\dagger = X^\top$.

$\qquad\qquad\square$

**Lemma 8** *For rank-1 matrix A, we have that $\|A\| = |\mathrm{tr}\,[A]|$.*

## D  PROOF OF THEOREM 2

In this section, we provide the proof of Theorem 2. We first recall the formal statement of Theorem 2.

**Theorem 2 (Decomposition Theorem)** *Assume that the excess risk has a unique minimizer with local sharpness factor $s$. Under Assumption 1 and Assumption 2, the following decomposition inequality holds:*

$$\mathcal{E}_{\mathcal{L}}(\hat{\boldsymbol{\theta}}^{(T)}; \mathcal{P}) \leq [4a]^s \frac{M_u}{m_u} \left[ \mathcal{E}_{\mathcal{L}}^v(\hat{\boldsymbol{\theta}}_v^{(T)}; \mathcal{P}) + \mathcal{E}_{\mathcal{L}}^b(\hat{\boldsymbol{\theta}}_b^{(T)}; \mathcal{P}) \right] + M_u \left[ \frac{4C}{\sqrt{T}} \right]^s + M_u \left[ \frac{4C'}{\sqrt{n}} \right]^s. \qquad (5)$$

**Proof:** Under Assumption 1 (upper bound), we have:

$$\mathcal{E}_{\mathcal{L}}(\hat{\boldsymbol{\theta}}^{(t)}; \mathcal{P}) = \frac{\mathcal{E}_{\mathcal{L}}(\hat{\boldsymbol{\theta}}^{(t)}; \mathcal{P})}{\|\hat{\boldsymbol{\theta}}^{(t)} - \hat{\boldsymbol{\theta}}^*\|^s} \|\hat{\boldsymbol{\theta}}^{(t)} - \hat{\boldsymbol{\theta}}^*\|^s$$

$$\leq \left[ M_u^{1/s} \|\hat{\boldsymbol{\theta}}^{(t)} - \hat{\boldsymbol{\theta}}^*\| \right]^s.$$

Combining Assumption 2 (DDC) and the lower bound in Assumption 1, we have

$$\|\hat{\boldsymbol{\theta}}^{(t)} - \hat{\boldsymbol{\theta}}^*\|$$

$$\leq a\|\hat{\boldsymbol{\theta}}_b^{(t)} - \hat{\boldsymbol{\theta}}_b^*\| + a\|\hat{\boldsymbol{\theta}}_v^{(t)} - \hat{\boldsymbol{\theta}}_v^*\| + \frac{C}{\sqrt{t}} + \frac{C'}{\sqrt{n}}$$

$$\leq [\frac{1}{m_u}]^{1/s} a \mathcal{E}_{\mathcal{L}}^{1/s}(\hat{\boldsymbol{\theta}}_b^{(t)}; \mathcal{P}_b) + [\frac{1}{m_u}]^{1/s} a \mathcal{E}_{\mathcal{L}}^{1/s}(\hat{\boldsymbol{\theta}}_v^{(t)}; \mathcal{P}_v) + \frac{C}{\sqrt{t}} + \frac{C'}{\sqrt{n}},$$

where we remark that Assumption 1 holds uniformly on distribution $\mathcal{P}_{\boldsymbol{y}|\boldsymbol{x}}$ and $\boldsymbol{\theta}$. Therefore, we have that

$$\mathcal{E}_{\mathcal{L}}(\hat{\boldsymbol{\theta}}^{(t)}; \mathcal{P}) \leq \left[ \frac{M_u^{1/s}}{m_u} a \mathcal{E}_{\mathcal{L}}^{1/s}(\hat{\boldsymbol{\theta}}_b^{(t)}; \mathcal{P}_b) + \frac{M_u^{1/s}}{m_u} a \mathcal{E}_{\mathcal{L}}^{1/s}(\hat{\boldsymbol{\theta}}_v^{(t)}; \mathcal{P}_v) + \frac{C}{\sqrt{t}} + \frac{C'}{\sqrt{n}} \right]^s$$

$$\leq 4^s \left[ \frac{M_u}{m_u} a^s \mathcal{E}_{\mathcal{L}}(\hat{\boldsymbol{\theta}}_b^{(t)}; \mathcal{P}_b) + \frac{M_u}{m_u} a^s \mathcal{E}_{\mathcal{L}}(\hat{\boldsymbol{\theta}}_v^{(t)}; \mathcal{P}_v) + \left[ M_u^{1/s} \frac{C}{\sqrt{t}} \right]^s + \left[ M_u^{1/s} \frac{C'}{\sqrt{n}} \right]^s \right]$$

$$\leq [4a]^s \frac{M_u}{m_u} \left[ \mathcal{E}_{\mathcal{L}}(\hat{\boldsymbol{\theta}}_b^{(t)}; \mathcal{P}_b) + \mathcal{E}_{\mathcal{L}}(\hat{\boldsymbol{\theta}}_v^{(t)}; \mathcal{P}_v) \right] + M_u \left[ \frac{4C}{\sqrt{t}} \right]^s + M_u \left[ \frac{4C'}{\sqrt{n}} \right]^s.$$

where we use the fact that $(a + b + c + d)^p \leq \max\{(4a)^p, (4b)^p, (4c)^p, (4d)^p\} \leq (4a)^p + (4b)^p + (4c)^p + (4d)^p$ when $a, b, c, d, p > 0$.

$\square$

# E    PROOF FOR DIAGONAL MATRIX RECOVERY

In this section, we prove Theorem 3 via proving the proceeding three lemmas separately.

## E.1    PROOF OF THEOREM 3

**Theorem 3 (Decomposition Theorem in Diagonal Matrix Recovery)** *In the diagonal matrix recovery settings, if we set the initialization $U_0 = \alpha I$ where $\alpha > 0$ is the initialization scale, then for any $0 < t < \infty$, with probability at least $1 - \delta$, we have:*

$$\mathcal{E}_{\mathcal{L}}(U_t; \mathcal{P}) \lesssim \left( \|X^\star\|_F^2 + dV^2 + d\alpha^4 \right) \sqrt{\frac{\log(d/\delta)}{n}} + \sum_{j=1}^{r} \frac{\sigma_j^4}{\sigma_j^2 + \alpha^4 e^{\Omega(\sigma_j t)}} + \frac{\alpha^2 d}{t} \tag{6}$$

$$+ \frac{dV^2(t+1)}{n} \log(n) \log\left( \frac{2dn}{\delta} \right) + d\alpha^2 + \log^2\left( \frac{1}{\alpha^2} \right) \frac{r\nu^2 \log(r/\delta)}{n}.$$

Note that although we do optimization based on $U$, we consider its square $X = UU^\top$ as the parameter since we require the solution is unique during the analysis. Note that we do not change the update rule (where GD is still taken on $U$). One can directly check that under Diagonal Matrix Recovery regimes, $s = 2$ and $M_u = m_u = 1$. Therefore, combining Lemma 4, Lemma 5 and Lemma 6 leads to Theorem 3.

## E.2    PROOF OF LEMMA 4

**Lemma 4** *Under the assumptions in Theorem 3, for any $0 \leq t < \infty$, with probability at least $1 - \delta$, we have*

$$\left\| U_t U_t^\top - X^\star \right\|_F \leq \left\| U_t^b U_t^{b,\top} - X^\star \right\|_F + \left\| U_t^v U_t^{v,\top} \right\|_F + \mathcal{O}\left( \log\left( \frac{1}{\alpha^2} \right) \sqrt{\frac{r\nu^2 \log(r/\delta)}{n}} \right). \tag{7}$$

**Proof:** Since all the measurement matrices $A^{(1)}, \cdots, A^{(n)}$ are diagonal, we can analyze the dynamic of each singular value separately.

For a given singular value $\sigma$, assume $a_i \sim \mathcal{N}(0,1)$, $\varepsilon_i \sim \text{subG}(\nu^2)$. Then we define $\sigma_1 = \frac{1}{n} \sum_{i=1}^{n} a_i^2 \sigma$, and $\sigma_2 = \frac{1}{n} \sum_{i=1}^{n} a_i \varepsilon_i$, denote $\sigma = \sigma_1 + \sigma_2$, and $\xi = \frac{1}{n} \sum_{i=1}^{n} a_i^2$. Denote $u^2(t)$, $u_b^2(t)$, $u_v^2(t)$ the dynamics of standard training, bias training, and variance training, respectively. Then our goal is to show that $|\sigma - u^2(t)| \leq |\sigma - u_b^2(t)| + u_v^2(t) + \text{small term}$ holds with high probability.

Before diving into the details, we first give the dynamics of $u^2(t)$, $u_b^2(t)$, $u_v^2(t)$. If $\sigma > 0$, we have

$$u^2(t) = \frac{\sigma \alpha^2 e^{2\sigma t}}{\sigma - \xi \alpha^2 + \xi \alpha^2 e^{2\sigma t}}.$$

$$u_b^2(t) = \frac{\sigma_1 \alpha^2 e^{2\sigma_1 t}}{\sigma_1 - \xi \alpha^2 + \xi \alpha^2 e^{2\sigma_1 t}}.$$

Furthermore, if $\sigma_2 > 0$, we have

$$u_v^2(t) = \frac{\sigma_2 \alpha^2 e^{2\sigma_2 t}}{\sigma_2 - \xi \alpha^2 + \xi \alpha^2 e^{2\sigma_2 t}}.$$

Otherwise, if $\sigma_2 < 0$ we have

$$u_v^2(t) = \frac{|\sigma_2| \alpha^2}{(|\sigma_2| + \xi \alpha^2) e^{2|\sigma_2|t} - \xi \alpha^2}.$$

If $\sigma^\star = 0$, we have

$$u_b^2(t) = \frac{\alpha^2}{\xi \alpha^2 t + 1}.$$

Now we turn to the proof of Lemma 4. If $\sigma = 0$, then $\sigma = \sigma_2$, which implies $u^2(t) = u_v^2(t)$, indicating that the conclusion holds.

Hence, we only consider the case that $\sigma \geq \sigma_r > 0$. By triangle inequality, it suffices to show

$$\left| u^2(t) - u_b^2(t) \right| = \mathcal{O}(|\sigma_2|).$$

Without loss of generality, we assume $\sigma > \sigma_1 > 0$. Note that

$$\left| u^2(t) - u_b^2(t) \right| = \left| \frac{\sigma \alpha^2 e^{2\sigma t}}{\sigma + \xi \alpha^2 e^{2\sigma t}} - \frac{\sigma_1 \alpha^2 e^{2\sigma_1 t}}{\sigma_1 + \xi \alpha^2 e^{2\sigma_1 t}} \right| + \mathcal{O}(|\sigma_2|)$$

$$= \frac{\alpha^2 \sigma \sigma_1 \left( e^{2\sigma t} - e^{2\sigma_1 t} \right)}{(\sigma + \xi \alpha^2 e^{2\sigma t})(\sigma_1 + \xi \alpha^2 e^{2\sigma_1 t})} + \mathcal{O}(|\sigma_2|).$$

Via integration-by-parts formula, we have

$$e^{2\sigma t} - e^{2\sigma_1 t} = 2\sigma_2 e^{2\sigma t} \int_0^t e^{-2\sigma_2 s} ds \leq 2\sigma_2 t e^{2\sigma t}.$$

Therefore, it suffices to show that

$$\frac{\sigma \sigma_1 \alpha^2 t e^{2\sigma t}}{(\sigma + \xi \alpha^2 e^{2\sigma t})(\sigma_1 + \xi \alpha^2 e^{2\sigma_1 t})} = \tilde{\mathcal{O}}(1).$$

We prove that there exists a universal constant $C$, such that

$$\frac{\sigma \sigma_1 \alpha^2 t e^{2\sigma t}}{(\sigma + \xi \alpha^2 e^{2\sigma t})(\sigma_1 + \xi \alpha^2 e^{2\sigma_1 t})} \leq C,$$

which is equivalent to show

$$\phi(t) = C \left( \sigma + \xi \alpha^2 e^{2\sigma t} \right) \left( \sigma_1 + \xi \alpha^2 e^{2\sigma_1 t} \right) - \sigma \sigma_1 \alpha^2 t e^{2\sigma t} \geq 0, \quad \forall 0 \leq t < \infty.$$

Note that $\phi(0) = C \left( \sigma + \xi \alpha^2 \right) \left( \sigma_1 + \xi \alpha^2 \right) \geq 0$, and its gradient is

$$\phi'(t) = 2C \xi^2 \alpha^4 (\sigma + \sigma_1) e^{2(\sigma + \sigma_1)t} + 2C \sigma \sigma_1 \xi \alpha^2 \left( e^{2\sigma t} + e^{2\sigma_1 t} \right) - \sigma \sigma_1 \alpha^2 e^{2\sigma t} - 2\sigma^2 \sigma \alpha^2 t e^{2\sigma t}.$$

By concentration of Chi-square distribution, we have $\xi \geq 1 - \sqrt{\frac{\log(1/\delta)}{n}} = \Omega(1)$ with probability at least $1 - \delta$, providing that $n \gtrsim \log\left(\frac{1}{\delta}\right)$. It suffices to show that $\phi'(t) \geq 0$, which is equivalent to

$$\psi(t) = C\alpha^2 e^{2\sigma_1 t} + C\sigma_1 - \sigma\sigma_1 t \geq 0.$$

It attains its minimum $\frac{\sigma}{2} + C\sigma_1 - \frac{\sigma}{2}\log\left(\frac{1}{2C\alpha^2}\right)$ at the point $t = \log\left(\frac{1}{2C\alpha^2}\right)/2\sigma_1$. Therefore, it suffices to set $C = \log\left(\frac{1}{\alpha^2}\right)$.

Overall, we have

$$\left|u^2(t) - u_b^2(t)\right| = \mathcal{O}\left(\log\frac{1}{\alpha^2}|\sigma_2|\right) \quad \text{if } \sigma^\star > 0;$$

$$\left|u^2(t) - u_b^2(t)\right| \leq u_v^2(t) \quad \text{if } \sigma^\star = 0.$$

Hence, with proper choice of probability, we have with probability at least $1 - \delta$, the following holds

$$\|X_t - X^\star\|_F \leq \|X_t^b - X^\star\|_F + \|X_t^v\|_F + \mathcal{O}\left(\log\left(\frac{1}{\alpha^2}\right)\sqrt{\frac{r\sigma^2 \log(r/\delta)}{n}}\right).$$

To summarize, the case is $\left(a = 1, C = 0, C' = \mathcal{O}\left(\log\left(\frac{1}{\alpha^2}\right)\sqrt{r\sigma^2 \log(r/\delta)}\right)\right)$-bounded (See DDC).

$\square$

### E.3 PROOF OF LEMMA 5

**Lemma 5** *For VER, under the assumptions in Theorem 3, with probability at least $1 - \delta$, we have*

$$\mathcal{E}_\mathcal{L}^v(U_t; \mathcal{P}) \lesssim \frac{dV^2(t+1)}{n}\log(n)\log\left(\frac{2dn}{\delta}\right) + d\alpha^2 + dV^2\sqrt{\frac{\log(2d/\delta)}{n}}. \tag{8}$$

**Proof:**

Similar to the proof in overparameterized linear regression regimes, we first split the excess risk in three componenets:

$$\mathcal{E}_\mathcal{L}(U_t; \mathcal{P}_v) = \underbrace{\mathcal{L}\left(u^2(t)\right) - \mathcal{L}_n\left(u^2(t)\right)}_{\text{(I)}} + \underbrace{\mathcal{L}_n\left(u^2(t)\right) - \mathcal{L}_n(0)}_{\text{(I)}} + \underbrace{\mathcal{L}_n(0) - \mathcal{L}(0)}_{\text{(III)}}. \tag{14}$$

**Part (I).** We bound the first component in Equation 13 using stability. We define $\sigma_2 = \frac{1}{n}\sum_{i=1}^n a_i\varepsilon_i$, $\sigma_2' = \frac{1}{n}\sum_{i=1}^{n-1} a_i\varepsilon_i + a_n'\varepsilon_n'$. Similarly, we define $\xi = \frac{1}{n}\sum_{i=1}^n a_i^2$, and $\xi' = \frac{1}{n}\sum_{i=1}^{n-1} a_i^2 + a_n'^2$.

*In the first case*, we assume both $\sigma_2, \sigma_2' > 0$, so that their difference can be bounded

$$|u_v^2(t) - u_v'^2(t)| \lesssim \frac{\left|\alpha^2\sigma_2\sigma_2'\left(e^{2\sigma_2 t} - e^{2\sigma_2' t}\right)\right| + \alpha^4\left|\xi'\sigma_2 e^{2\sigma_2 t} - \xi\sigma_2' e^{2\sigma_2' t}\right| + \alpha^4(\sigma_2\xi' - \sigma_2'\xi)e^{2(\sigma_2 + \sigma_2')t}}{\left(\sigma_2 - \xi\alpha^2 + \xi\alpha^2 e^{2\sigma_2 t}\right)\left(\sigma_2' - \xi'\alpha^2 + \xi'\alpha^2 e^{2\sigma_2' t}\right)}.$$

WLOG, we assume $\sigma_2 > \sigma_2'$. Then, similar to the proof in Lemma 4, we have

$$\frac{\left|\alpha^2\sigma_2\sigma_2'\left(e^{2\sigma_2 t} - e^{2\sigma_2' t}\right)\right|}{\left(\sigma_2 - \xi\alpha^2 + \xi\alpha^2 e^{2\sigma_2 t}\right)\left(\sigma_2' - \xi'\alpha^2 + \xi'\alpha^2 e^{2\sigma_2' t}\right)} = \mathcal{O}\left(|\sigma_2 - \sigma_2'|t\right).$$

The second term is dominated by the first term. Hence, we only need to handle the third term. Note that

$$|\sigma_2\xi' - \sigma_2'\xi| = |\sigma_2(\xi' - \xi) + \xi(\sigma_2 - \sigma_2')| \leq \frac{|\sigma_2|}{n}\left|a_n^2 - a_n'^2\right| + \left|\frac{\xi}{n}\left(a_n\varepsilon_n - a_n'\varepsilon_n'\right)\right|,$$

we have

$$\frac{\alpha^4(\sigma_2\xi' - \sigma_2'\xi)e^{2(\sigma_2 + \sigma_2')t}}{\left(\sigma_2 - \xi\alpha^2 + \xi\alpha^2 e^{2\sigma_2 t}\right)\left(\sigma_2' - \xi'\alpha^2 + \xi'\alpha^2 e^{2\sigma_2' t}\right)} \leq \frac{|\sigma_2\xi' - \sigma_2'\xi|}{\xi\xi'}$$

$$\lesssim \frac{|\sigma_2|}{n}\left|a_n^2 - a_n'^2\right| + \left|\frac{\xi}{n}\left(a_n\varepsilon_n - a_n'\varepsilon_n'\right)\right|.$$

Combined the above three parts, we finally have

$$|u_v^2(t) - u_v'^2(t)| = \mathcal{O}\left(\frac{V(t+1)}{n}\right).$$

*In the second case*, we assume both $\sigma_2, \sigma_2' < 0$. Since both of them would decrease to 0, we have $|u_b^2(t) - u_b'^2(t)| \leq \alpha^2$ where $\alpha = \mathcal{O}(\frac{1}{n^{1/4}})$.

*In the last case*, without loss of generality, we assume $\sigma_2 < 0 < \sigma_2'$. Since $u_v^2(t)$ decreases to 0, and $u'^2_v(t)$ increases to $\sigma_2'$, we have $|u_b^2(t) - u_b'^2(t)| \leq \sigma_2' \leq |\sigma_2' - \sigma_2|$.

Overall, we have

$$|u_v^2(t) - u_v'^2(t)| = \mathcal{O}\left(\frac{V(t+1)}{n}\right).$$

Besides, since the loss is $\mathcal{O}(V)$-Lipschitz:

$$\|\nabla_{u^2}\ell(u)\| = 2\left(au^2 - \varepsilon\right)a \lesssim V.$$

where $u^2$ is bounded upper bounded by $\mathcal{O}(V)$ during the training analysis due to a small initialization. Then the total stability is

$$|\ell(u_v^2(t)) - \ell(u_v'^2(t))| \leq \frac{V^2(t+1)}{n}.$$

Summation over the dimension $d$, the whole loss $\ell_d(u_v^2(t))$ has stability

$$|\ell_d(U_v^2(t)) - \ell_d(U_v'^2(t))| \leq \frac{dV^2(t+1)}{n}.$$

By Proposition 1, we have that

$$\mathcal{L}(\hat{\theta}_v^{(t)}; \mathcal{P}) - \mathcal{L}(\hat{\theta}_v^{(t)}; \mathcal{P}^n) \lesssim \frac{dV^2(t+1)}{n}\log(n)\log(2dn/\delta) + \frac{\sqrt{\log(1/\delta)}}{\sqrt{n}}.$$

**Part (II).** We next calculate the second term. Take the explicit formula into the equation, we have

$$\mathcal{L}_n\left(u^2(t)\right) - \mathcal{L}_n(0) = \frac{1}{n}\sum_{i=1}^{n}\left(a_i^2 u_v^4(t) - 2a_i\varepsilon_i u_v^2(t)\right).$$

If $\sigma_2 = \frac{1}{n}\sum_{i=1}^{n} a_i\varepsilon_i > 0$, we have

$$\frac{1}{n}\sum_{i=1}^{n}\left(a_i^2 u_v^4(t) - 2a_i\varepsilon_i u_v^2(t)\right) = -\frac{(\sigma_2 - \xi\alpha^2)\sigma_2^2\alpha^2 e^{2\sigma_2 t}}{\left(\sigma_2 - \xi\alpha^2 + \xi\alpha^2 e^{2\sigma_2 t}\right)^2} \leq 0.$$

If $\sigma_2 < 0$, we have

$$\frac{1}{n}\sum_{i=1}^{n}\left(a_i^2 u_v^4(t) - 2a_i\varepsilon_i u_v^2(t)\right) = \xi\frac{\sigma_2^2\alpha^4}{\left((|\sigma_2| + \xi\alpha^2)e^{2|\sigma_2|t} - \xi\alpha^2\right)^2} + \frac{2|\sigma_2|^2\alpha^2}{(|\sigma_2| + \xi\alpha^2)e^{2|\sigma_2|t} - \xi\alpha^2}$$

$$\leq \xi\alpha^4 + 2|\sigma_2|\alpha^2 \lesssim \alpha^2\sqrt{\frac{\nu^2}{n}}.$$

In conclusion, we have $\mathcal{L}_n\left(u^2(t)\right) - \mathcal{L}_n(0) \lesssim \alpha^2\sqrt{\frac{\nu^2}{n}}$.

**Part (III).** For the third term, we just need one single concentration inequality. Note that for the variance training, the loss function is of the order $\mathcal{O}\left(V^2\right)$. Hence, via the Hoeffding inequality, with probability at least $1 - \delta$, we have

$$\mathcal{L}_n(0) - \mathcal{L}(0) \lesssim V\sqrt{\frac{\log(1/\delta)}{n}}.$$

Combining the three terms leads to Lemma 5.

$\square$

### E.4 PROOF OF LEMMA 6

**Lemma 6** *For BER, under the assumptions in Theorem 3, with probability at least $1 - \delta$, we have*

$$\mathcal{E}_{\mathcal{L}}^b(U_t; \mathcal{P}) \lesssim \left( \|X^\star\|_F^2 + d\alpha^4 \right) \sqrt{\frac{\log\left(d/\delta\right)}{n}} + \sum_{j=1}^r \frac{\sigma_j^4}{\sigma_j^2 + \alpha^4 e^{\Omega(\sigma_j t)}} + \frac{\alpha^2 d}{t}. \qquad (9)$$

**Proof:**

Similar to linear regression, we decompose the excess risk into two parts:

$$\mathbb{E}\left[ a^2 \left( u_b^2(t) - \sigma \right)^2 \right] = \mathbb{E}\left[ a^2 - \frac{1}{n}\sum_{i=1}^n a_i^2 \right] \left( u_b^2(t) - \sigma \right)^2 + \frac{1}{n}\sum_{i=1}^n a_i^2 \left( u_b^2(t) - \sigma \right)^2.$$

We use the uniform convergence to bound the first term. Assume that $\sigma > 0$, we have

$$
\begin{aligned}
\sigma - u_b^2(t) &= \sigma - \frac{\sigma_1 \alpha^2 e^{2\sigma_1 t}}{\sigma_1 - \xi\alpha^2 + \xi\alpha^2 e^{2\sigma_1 t}} \\
&= \frac{\sigma\sigma_1 - \xi\alpha^2\sigma + \alpha^2 e^{2\sigma_1 t}\left(\xi\sigma - \sigma\right)}{\sigma_1 - \xi\alpha^2 + \xi\alpha^2 e^{2\sigma_1 t}} \\
&= \frac{\sigma\sigma_1 - \xi\alpha^2\sigma}{\sigma_1 - \xi\alpha^2 + \xi\alpha^2 e^{2\sigma_1 t}} \\
&= \frac{\sigma^2 - \alpha^2\sigma}{\sigma - \alpha^2 + \alpha^2 e^{2\xi\sigma t}} \\
&\le \sigma(1 - \alpha^2) \le \sigma.
\end{aligned}
$$

Now assume $\sigma = 0$, we have

$$u_b^2(t) = \frac{\alpha^2}{\xi\alpha^2 t + 1} \le \alpha^2.$$

The second term refers to the training loss, if $\sigma > 0$, we have

$$\frac{1}{n}\sum_{i=1}^n a_i^2 \left( u_b^2(t) - \sigma \right)^2 = \xi\frac{\left(\sigma^2 - \alpha^2\sigma\right)^2}{\left(\sigma - \alpha^2 + \alpha^2 e^{2\xi\sigma t}\right)^2} \lesssim \frac{\sigma^4}{\sigma^2 + \alpha^4 e^{\Omega(\sigma t)}}.$$

If $\sigma = 0$, we have

$$\frac{1}{n}\sum_{i=1}^n a_i^2 \left( u_b^2(t) - \sigma \right)^2 = \xi\frac{\alpha^4}{\left(\xi\alpha^2 t + 1\right)^2} \lesssim \frac{\alpha^4}{1 + \alpha^4 t^2}.$$

Finally, by assigning probability properly, we have

$$\mathcal{E}_{\mathcal{L}}(U_t; \mathcal{P}_b) \lesssim \left( \|X^\star\|_F^2 + d\alpha^4 \right) \sqrt{\frac{\log\left(d/\delta\right)}{n}} + \sum_{j=1}^r \frac{\sigma_j^4}{\sigma_j^2 + \alpha^4 e^{\Omega(\sigma_j t)}} + \frac{\alpha^2 d}{t}.$$

$\qquad\qquad\qquad\qquad\qquad\qquad\qquad\qquad\qquad\qquad\qquad\qquad\qquad\qquad\qquad\qquad\qquad\qquad\quad\square$

## F  ADDITIONAL DISCUSSION

In this section, we discuss more the details in the main text. We first show a case where our bound outperforms previous stability-based bound under linear regimes. We then validate the rationality of applying stability-based bound by showing that the generalization gap (under variance regime) increases with time. We next provide some motivation behind Dynamics Decomposition Condition by showing that it indeed holds as the beginning of the training phase. We finally introduce a related version of Dynamics Decomposition Condition which does not require the information about $\theta^*$, $\theta_v^*$, and $\theta^*$.

### F.1 COMPARISON TO PREVIOUS STABILITY-BASED BOUNDS UNDER LINEAR REGIMES

One of the shortcomings of the stability-based bound is its failure with a large time $T$. Our decomposition framework can aid in reducing the failure at a large time $T$. For example, when considering a large time $T = n^{3/4}$ with $\lambda = 1$, the newly-proposed decomposition bound is

$$\mathcal{O}(n^{-1/4}[V + B]^2).$$

As a comparison, the original stability-based bound is

$$\mathcal{O}(n^{-1/4}[V + B']^2).$$

We next show that $B < B'$ holds with a large signal-to-noise ratio.

We write the above formula as

$$\|\hat{\boldsymbol{\theta}}_v^{(t)}\| = B < B' = \|\hat{\boldsymbol{\theta}}^{(t)}\|.$$

From the formulation of the trained parameter, we have $\hat{\boldsymbol{\theta}}_v^{(t)} = [I - [I - \frac{\lambda}{n}X^\top X]^t]X^\dagger \boldsymbol{\epsilon}$ and $\hat{\boldsymbol{\theta}}^{(t)} = [I - [I - \frac{\lambda}{n}X^\top X]^t]X^\dagger Y = [I - [I - \frac{\lambda}{n}X^\top X]^t]\boldsymbol{\theta}^* + \hat{\boldsymbol{\theta}}_v^{(t)}$. Therefore, it suffices to show that $\|[I - [I - \frac{\lambda}{n}X^\top X]^t]\boldsymbol{\theta}^*\| > 2\|\hat{\boldsymbol{\theta}}_v^{(t)}\|$.

Note that the first part is only related to signal $\boldsymbol{\theta}^*$ and the second part is related to noise $\boldsymbol{\epsilon}$ (by applying concentration, the second part is closely related to the noise level). Therefore, informally, with a large signal-to-noise ratio, the last equation $\|[I - [I - \frac{\lambda}{n}X^\top X]^t]\boldsymbol{\theta}^*\| > 2\|\hat{\boldsymbol{\theta}}_v^{(t)}\|$ holds. To summary, it holds that $B < B'$ and our bound outperforms the original stability-based bounds.

### F.2 THE GENERALIZATION GAP UNDER VARIANCE INCREASES WITH TIME

In this part, we will prove that the (variance regime) generalization gap indeed increases with time, which validates that applying stability-based bound (which also increases with time) is rational.

We provide the following Lemma 9 which validates the above statement.

**Lemma 9** *Let $\Delta_v(t) = \mathcal{L}(\hat{\boldsymbol{\theta}}_v^{(t)}; \mathcal{P}) - \mathcal{L}(\hat{\boldsymbol{\theta}}_v^{(t)}; \mathcal{P}^n)$ be the generalization gap under variance regime. Assume that $\Sigma_{\boldsymbol{x}} = I$. Then under linear regression settings as in Section 3, we have*

$$\frac{\partial}{\partial t}\Delta_v(t) \geq 0.$$

**Proof:** Note that the population loss is calculated as follows by Equation C.1

$$\mathcal{L}(\hat{\boldsymbol{\theta}}_v^{(t)}; \mathcal{P}) = \left[\hat{\boldsymbol{\theta}}_v^{(t)}\right]^\top \Sigma_x \left[\hat{\boldsymbol{\theta}}_v^{(t)}\right] + \mathbb{E}\epsilon^2.$$

And the training loss is calculated as follows:

$$\begin{aligned}
\mathcal{L}(\hat{\boldsymbol{\theta}}_v^{(t)}; \mathcal{P}^n) &= \frac{1}{n}\|E - X\hat{\boldsymbol{\theta}}_v^{(t)}\|^2 \\
&= \frac{1}{n}\|X\hat{\boldsymbol{\theta}}_v^{(t)}\|^2 - \frac{2}{n}E^\top \left[X\hat{\boldsymbol{\theta}}_v^{(t)}\right] + \frac{1}{n}\|E\|^2 \\
&= \left[\hat{\boldsymbol{\theta}}_v^{(t)}\right]^\top \left[\frac{1}{n}X^\top X\right]\left[\hat{\boldsymbol{\theta}}_v^{(t)}\right] - \frac{2}{n}E^\top X\hat{\boldsymbol{\theta}}_v^{(t)} + \frac{1}{n}\|E\|^2
\end{aligned}.$$

Note that by Lemma 7, we have

$$\hat{\boldsymbol{\theta}}_v^{(t)} = \left[I - \left[I - \frac{\lambda}{n}X^\top X\right]^t\right][X^\dagger E].$$

Therefore, we have:

$$\Delta_v(t) = \mathcal{L}(\hat{\boldsymbol{\theta}}_v^{(t)}; \mathcal{P}) - \mathcal{L}(\hat{\boldsymbol{\theta}}_v^{(t)}; \mathcal{P}^n)$$

$$= \left[\hat{\boldsymbol{\theta}}_v^{(t)}\right]^\top \Sigma_x \left[\hat{\boldsymbol{\theta}}_v^{(t)}\right] + \mathbb{E}\epsilon^2 - \left[\hat{\boldsymbol{\theta}}_v^{(t)}\right]^\top \left[\frac{1}{n}X^\top X\right]\left[\hat{\boldsymbol{\theta}}_v^{(t)}\right] + \frac{2}{n}E^\top X\hat{\boldsymbol{\theta}}_v^{(t)} - \frac{1}{n}\|E\|^2$$

$$= \left[\hat{\boldsymbol{\theta}}_v^{(t)}\right]^\top \left[\Sigma_x - \frac{1}{n}X^\top X\right]\left[\hat{\boldsymbol{\theta}}_v^{(t)}\right] + \frac{2}{n}E^\top X\hat{\boldsymbol{\theta}}_v^{(t)} + \mathbb{E}\epsilon^2 - \frac{1}{n}\|E\|^2$$

$$= E^\top \left[X^\dagger\right]^\top \left[I - \left[I - \frac{\lambda}{n}X^\top X\right]^t\right]\left[\Sigma_x - \frac{1}{n}X^\top X\right]\left[I - \left[I - \frac{\lambda}{n}X^\top X\right]^t\right]\left[X^\dagger E\right]$$

$$+ \frac{2}{n}E^\top X\left[I - \left[I - \frac{\lambda}{n}X^\top X\right]^t\right]\left[X^\dagger E\right] + \mathbb{E}\epsilon^2 - \frac{1}{n}\|E\|^2.$$

By omitting the terms in $\Delta_v(t)$ that is uncorrelated to $t$, we have (denote as $\bar{\Delta}_v(t)$):

$$\bar{\Delta}_v(t) = E^\top \left[X^\dagger\right]^\top \left[I - \frac{\lambda}{n}X^\top X\right]^t \left[\Sigma_x - \frac{1}{n}X^\top X\right]\left[I - \frac{\lambda}{n}X^\top X\right]^t \left[X^\dagger E\right]$$

$$- 2E^\top \left[X^\dagger\right]^\top \left[\Sigma_x - \frac{1}{n}X^\top X\right]\left[I - \frac{\lambda}{n}X^\top X\right]^t \left[X^\dagger E\right]$$

$$- \frac{2}{n}E^\top X\left[I - \frac{\lambda}{n}X^\top X\right]^t \left[X^\dagger E\right]$$

$$= E^\top \left[X^\dagger\right]^\top \left[I - \frac{\lambda}{n}X^\top X\right]^t \left[\Sigma_x - \frac{1}{n}X^\top X\right]\left[I - \frac{\lambda}{n}X^\top X\right]^t \left[X^\dagger E\right]$$

$$- 2E^\top \left[X^\dagger\right]^\top \left[\Sigma_x\right]\left[I - \frac{\lambda}{n}X^\top X\right]^t \left[X^\dagger E\right]$$

$$= E^\top \left[X^\dagger\right]^\top \left[\left[I - \frac{\lambda}{n}X^\top X\right]^t \left[\Sigma_x - \frac{1}{n}X^\top X\right] - 2\Sigma_x\right]\left[I - \frac{\lambda}{n}X^\top X\right]^t \left[X^\dagger E\right].$$

where we use the fact that $\left[X^\dagger\right]^\top X^\top X = X$.

We next focus on the matrix

$$M_a \triangleq \left[X^\dagger\right]^\top \left[\left[I - \frac{\lambda}{n}X^\top X\right]^t \left[I - \frac{1}{n}X^\top X\right] - 2I\right]\left[I - \frac{\lambda}{n}X^\top X\right]^t X^\dagger,$$

where we plug in $\Sigma_{\boldsymbol{x}} = I$ by assumption. Denote the SVD of $\frac{1}{n}X^\top X$ as $U^\top\Sigma U$ where the $i$-th eigenvalue is $\sigma_i$. Then when $\sigma_i = 0$, the $i - th$ eigenvalue of $M_a$ is also equal to zero (due to the persudo inverse $X^\dagger$). When $\sigma_i > 0$, the $i - th$ eigenvalue of $M_a$ can be derived as

$$\sigma_i(t; M_a) = \frac{\left[(1 - \lambda\sigma_i)^t(1 - \sigma_i) - 2\right](1 - \lambda\sigma_i)^t}{\sigma_i}.$$

Its derivation is then

$$\frac{\partial}{\partial t}\sigma_i(t; M_a) = \frac{2(1 - \lambda\sigma_i)^t \log(1 - \lambda\sigma_i)}{\sigma_i}\left[(1 - \sigma_i)(1 - \lambda\sigma_i)^t - 1\right] \geq 0, \tag{15}$$

since $\log(1 - \lambda\sigma_i) \leq 0$ and $(1 - \sigma_i)(1 - \lambda\sigma_i)^t - 1 \leq 0$. Therefore, the derivation of each eigenvalue is larger than zero.

We rewrite $\bar{\Delta}_v(t) = z^\top \Sigma_{Ma} z^\top$ where $z^\top = UE$ is a vector independent of time t and $\Sigma_{Ma}$ is the diagonal matrix with diagonal $\sigma_i(M_a)$. As a result, we have

$$\frac{\partial}{\partial t}\bar{\Delta}_v(t) = z^\top (\frac{\partial}{\partial t}\Sigma_{Ma})z^\top \geq 0,$$

since $\frac{\partial}{\partial t}\Sigma_{Ma} \succeq 0$ according to Equation 15.

Note that $\bar{\Delta}_v(t)$ and $\Delta_v(t)$ only differ with a term independent of time, we have

$$\frac{\partial}{\partial t}\Delta_v(t) = \frac{\partial}{\partial t}\bar{\Delta}_v(t) \geq 0.$$

$\square$

### F.3 MOTIVATION BEHIND DYNAMICS DECOMPOSITION CONDITION

For a dataset $\mathcal{S} = \{(X_i, y_i)\}_{i=1}^n$, where $y_i = f_{\boldsymbol{\theta}^\star}(X_i) + \varepsilon_i$, and $\boldsymbol{\theta}^\star \in \boldsymbol{\theta}$. We choose the $\ell_2$-loss to solve this problem $\mathcal{L}(\boldsymbol{\theta}) = \frac{1}{2n}\sum_{i=1}^n (f_{\boldsymbol{\theta}}(X_i) - y_i)^2$. Then we can show that the Dynamics Decomposition Condition holds at the beginning phase if we initialize $\boldsymbol{\theta}_v^{(0)} = 0$, and assume $f_0(X) = 0, \forall X \in \mathsf{supp}\{\mathcal{P}\}$. For sufficient small $t > 0$, the dynamics are well approximated by

$$\hat{\boldsymbol{\theta}}^{(t)} = \boldsymbol{\theta}^{(0)} - \frac{1}{n}\sum_{i=1}^n (f_{\boldsymbol{\theta}^{(0)}}(X_i) - y_i)\nabla f_{\boldsymbol{\theta}^{(0)}}(X_i)t;$$

$$\hat{\boldsymbol{\theta}}_b^{(t)} = \boldsymbol{\theta}_b^{(0)} - \frac{1}{n}\sum_{i=1}^n \left(f_{\boldsymbol{\theta}_b^{(0)}}(X_i) - f_{\boldsymbol{\theta}^\star}(X_i)\right)\nabla f_{\boldsymbol{\theta}_b^{(0)}}(X_i)t; \qquad (16)$$

$$\hat{\boldsymbol{\theta}}_v^{(t)} = \boldsymbol{\theta}_v^{(0)} - \frac{1}{n}\sum_{i=1}^n \left(f_{\boldsymbol{\theta}_v^{(0)}}(X_i) - \varepsilon_i\right)\nabla f_{\boldsymbol{\theta}_v^{(0)}}(X_i)t.$$

Suppose $\boldsymbol{\theta}^{(0)} = \boldsymbol{\theta}_b^{(0)}$, then based on Taylor expansion, we have the approximation

$$\hat{\boldsymbol{\theta}}^{(t)} \approx \hat{\boldsymbol{\theta}}_b^{(t)} + \hat{\boldsymbol{\theta}}_v^{(t)} + \frac{1}{n}\sum_{i=1}^n \varepsilon_i\nabla^2 f_{\boldsymbol{\theta}^{(0)}}(X_i)\left(\boldsymbol{\theta}_v^{(0)} - \boldsymbol{\theta}^{(0)}\right)t \approx \hat{\boldsymbol{\theta}}_b^{(t)} + \hat{\boldsymbol{\theta}}_v^{(t)}. \qquad (17)$$

Therefore, the Dynamics Decomposition Condition holds at least at the beginning phase. Moreover, we discuss in the main text that for a branch of learning tasks, such as linear regression and matrix recovery, these Dynamics Decomposition Condition uniformly holds for arbitrary $0 \leq t < \infty$.

### F.4 RELAXED VERSION OF DDC

One may notice that there exist $\boldsymbol{\theta}^*, \boldsymbol{\theta}_v^*, \boldsymbol{\theta}_b^*$ in DDC (See Assumption 2), which is the optimal solution to the corresponding population loss. They are sometimes unavailable in practice and even have no explicit solution. For more convenient analysis, we propose a relaxed version of DDC in Lemma 10.

**Lemma 10 (Relaxed DDC)** *Let the loss be $\ell_2$ loss where $\ell(\boldsymbol{y}, \hat{\boldsymbol{y}}) = (\boldsymbol{y} - \hat{\boldsymbol{y}})^2$. We assume the predicting model $\hat{y} = f_{\hat{\boldsymbol{\theta}}}(\boldsymbol{x})$ is unique with respect to $\hat{\boldsymbol{\theta}}$ (when $\boldsymbol{\theta}_i \neq \boldsymbol{\theta}_j$, $f_{\boldsymbol{\theta}_i} \neq f_{\boldsymbol{\theta}_j}$). Besides, assume $f_{\boldsymbol{\theta}=0}(\boldsymbol{x}) = 0$ for all $\boldsymbol{x}$. For the ground truth, we assume an additive noise, namely, $\boldsymbol{y} = f_{\boldsymbol{\theta}^*}(\boldsymbol{x}) + \epsilon$ where $\epsilon$ is independent of $\boldsymbol{x}$. Then the following Equation 18 suffices to prove $(\max\{a, 1\}, C, C')$-boundedness for DDC (See Assumption 2).*

$$\|\hat{\boldsymbol{\theta}}^{(t)} - \hat{\boldsymbol{\theta}}_b^{(t)}\| \leq a\|\hat{\boldsymbol{\theta}}_v^{(t)}\| + \frac{C}{\sqrt{t}} + \frac{C'}{\sqrt{n}}. \qquad (18)$$

Based on Lemma 10, one can verify DDC without knowing the optimal solution parameter $\boldsymbol{\theta}^*, \boldsymbol{\theta}_v^*, \boldsymbol{\theta}_b^*$. Besides, since we usually have the iteration equation (the relationship between $\hat{\boldsymbol{\theta}}^{(t+1)}$ and $\hat{\boldsymbol{\theta}}^{(t)}$) given an algorithm, it is possible to verify DDC based on the iteration forms. Note that we may not directly verify the condition in practice based on the relaxed DDC if we do not have the explicit split on signal $\mathbb{E}[\boldsymbol{y}|\boldsymbol{x}]$ and noise $\boldsymbol{y} - \mathbb{E}[\boldsymbol{y}|\boldsymbol{x}]$.

