# OpenReview forum: "Towards Understanding Generalization via Decomposing Excess Risk Dynamics"
_ICLR.cc/2022/Conference — ICLR 2022 Poster_

### Official Review · Reviewer_PEVU · 2021-10-18

**Correctness:** 3
**Technical Novelty And Significance:** 2
**Empirical Novelty And Significance:** 2
**Recommendation:** 5
**Confidence:** 4

**Details Of Ethics Concerns:**

There are no ethics concerns.

**Main Review:**

The idea of decomposing excess risk dynamics seems to be novel and interesting. However, there are some issues.

- The advantage of the results is not quite clear. For example, the authors indicated that Thm 1 outperforms the existing stability analysis by replacing $B'$ with $B$. However, this improvement is minor as both two quantities are unknown to us and it is not clear which one would be significantly larger. While the authors mentioned that $B$ is independent of $\|\theta^* \|$. This seems not important since the excess risk bound already involves $\|\theta^* \|$.

- The decomposition theorem requires Assumption 1 and Assumption 2, which may be restrictive. While the authors mentioned that they hold for some specific cases, in general cases these conditions are difficult to check in practice and may not hold.

- Assumption 1 on the uniform bound is imposed for $\theta$ and $P$. However, in the proof of Thm 2, the authors also useed the uniform bound assumption for $(\theta^{(t)}_b,P_b)$ and $(\theta^{(t)}_v,P_v)$. The authors should indicate these assumptions clearly in Assumption 1.

- The stability approach has a drawback of not yielding good bounds for nonconvex problems, while the uniform convergence approach has a drawback of yielding dimension-dependent bounds. The authors proposed to use stability approach and uniform convergence approach to tackle the noise and signal separately. The theoretical results therefore may inherit the drawbacks of both stability and uniform convergence: not appealing for nonconvex problems and yielding dimension-dependent bounds.

- The analysis seems not rigorous. In particular, the authors proved in page 19 that $\ell(\theta^{(t)},P_v^n)\leq \ell(\theta^*,P_v^n)$. This is not quite intuitive since this result holds for all $t$. In particular, in the first few iterations the performance of gradient descent may not be good and it is unlikely that they have smaller training errors than $\theta^*$. Furthermore, the notation $\epsilon$ in page 19 is not clear. It should be a vector according to the norm there. This causes confusion since $\epsilon$ is a real-valued random variable in Thm 1.

Typos:
- page 17: $E[\epsilon x^\top\theta^{(t)}]$ should be $-2E[\epsilon x^\top\theta^{(t)}]$
- page 18: $x_i^\top\theta_v+\theta_b$ should be $x_i^\top(\theta_v+\theta_b)$
- proof of Lemma 7: some $\theta$ should be $\hat{\theta}$. $\theta_0$ should be $\theta^{(0)}$
- page 23: $M_u^{1/s}/m_u$ should be $(M_u/m_u)^{\frac{1}{s}}$. There is also a missing factor of $m_u^{1/s}$

**Summary Of The Paper:**

This paper studies the generalization performance of learning algorithms. The basic idea is to decompose the training dynamics into noise and signal components, and then consider separately the behavior of models trained with noise and signals. The authors then proposed to use the stability approach to tackle the noise components, and the uniform convergence approach to tackle the signal part. The authors consider two specific algorithms: overparameterized linear regression and diagonal matrix recovery. Experimental results are also reported.

**Summary Of The Review:**

The idea seems to be novel. However, the advantage of the decomposing excess risk dynamics is not well justified. There are also some issues on the correctness of the deduction.

---

> ### Author Response · Authors · 2021-11-16
> **Clarifications**
>
> Thank you for your fair assessment of the novelty of our work.
> Below, we try to do our best to address your questions adequately and ensure that the deduction is correct.
> We encourage you to read the “General Response” section first and return here for further discussion.
>
> > The advantage of the results is not quite clear. [...] the comparison between B and B' [...] This seems not important since the excess risk bound already involves $\theta^*$.
>
> We apologize for not making all the statements crystal clear. Firstly, the merits of the decomposition framework are more conceptual than technological (see [General Response: Contribution of the paper]), implying that our bound outperforms original stability-based bounds in the sense that it accords better with the properties of neural networks. Secondly, we provide a detailed analysis comparing $B$ and $B^\prime$ in the [General Response: Comparison with original stability-based bounds]. Thirdly, we note that the dominated term in the decomposition bound does not include $\Vert \theta^* \Vert$ for a large time (e.g., $t = n^{3/4} > n^{1/2}$). We will add these discussions with more details in the next version. We thank the reviewer for giving us the opportunity to polish our work.
>
> > The decomposition theorem requires Assumption 1 and Assumption 2, which may be restrictive.
>
> The decomposition theorem provided in the current edition is not the only approach to reaching the decomposition goal, as argued in the main text.
> For example, we conduct several experiments on neural networks to illustrate that neural networks exhibit the decomposition phenomenon, although the assumptions do not explicitly hold. We leave the theoretical discussion of the decomposition theorem for future work.
>
> > Assumption 1 is imposed for $\theta$ and $P$, [...], in the proof of Thm2 [...].
>
> The statements here are correct since Assumption 1 is assumed to hold for any $P_{y|x}$ and any $\theta$. Therefore, assumption 1 is already enough for the proof of Thm2 (which can be viewed as a special $P_{y|x}$). Sorry for the confusion, and we will make it clearer in the next version.
>
>
> > The stability approach has a drawback of not yielding good bounds for non-convex problems, while the uniform convergence approach has a drawback of yielding dimension-dependent bounds. [...] The theoretical results therefore may inherit the drawbacks of both stability and uniform convergence: not appealing for non-convex problems and yielding dimension-dependent bounds.
>
> As proven in the overparameterized linear regression and diagonal matrix recovery, a proper decomposition benefits from inheriting the merits of both stability-based bound and uniform convergence bound.
> For example, the decomposition framework can be employed in non-convex regimes (as we can apply uniform convergence to deal with the non-convex component) and avoid the dimension-dependent issues (uniform convergence can avoid dimension-dependent bounds by norm-dependent bounds, as used in the main text.)
>
>
> > The analysis seems not rigorous. [...] in the first few iterations the performance of gradient descent may not be good and it is unlikely that they have smaller training errors than $\theta^*$.
>
> The analysis is indeed rigorous. Note that by assuming the initialization $\theta^{(0)} = 0$ (which is standard in overparameterized linear regression), the formulation in the first few iterations can be guaranteed (as the proof shows). For general cases, we have a 1/\sqrt{t} corrections in case that the conclusion does not hold at the beginning phase.
>
>
> > Furthermore, the notation $\epsilon$ in page 19 is not clear. It should be a vector according to the norm there. [...] and other typos [...].
>
> We apologize for some typos that have slipped through. We make sure to fix them all for the final version.
>
>
> > The idea seems to be novel. However, the advantage of the decomposing excess risk dynamics is not well justified.
>
> Thanks for your affirmation on the novelty of the paper. Compared to the original stability-based bound, the decomposition framework accords better with the neural networks since the framework is motivated by the observations in deep learning.
> Although it is a very early step towards explaining generalization in neural networks, we believe it brings some insights into the stability-based bounds and deep learning community.
>
>
> > There are also some issues on the correctness of the deduction.
>
> We believe that the statements in the proof are rigorous based on the above discussion. Thanks for the constructive comments for us to improve the paper.
>
> Once again, we thank the reviewer for their constructive comments that will help us shape discussion points, and we hope they will agree with the value of our contributions and the rigor of the proof in light of our responses.
> We are eager to engage in further discussions to clear out any confusion.

---

### Official Review · Reviewer_HZ9d · 2021-11-01

**Correctness:** 4
**Technical Novelty And Significance:** 2
**Empirical Novelty And Significance:** Not applicable
**Recommendation:** 5
**Confidence:** 3

**Main Review:**

This paper is written clearly and easy to follow. Under the linear regression setting, the decomposition makes a lot of sense because the dynamics can be split exactly into two parts. Compared with directly applying the stability arguments, the decomposition removes the reliance on the norm of the parameter $\theta^*$.

The first concern is, how better is Theorem 1 compared to directly using uniform convergence to characterize the excess risk of linear regression, especially under the high signal-noise-ration scenario. It is argued that Theorem 1 is better than directly applying stability results under the high signal-noise-ration scenario, which makes sense. But I wonder if the proposed decomposition really improves the state-of-the-art risk bounds. Some added discussion may help.

The next concern is the applicability of this decomposition. For the decomposition to make sense in general regimes, some very strong assumptions are made. For example, Assumption 2 directly assumes the training dynamics can be decomposed into two parts, which I believe can hardly happen in most cases.

I also have concerns with whether this decomposition is meaningful for modern deep learning tasks. In my opinion, this decomposition is meaningful typically under a setting where the 'bias training' is easy and the noise plays a nonnegligible role in the training dynamics. In modern deep learning tasks, usually, the 'bias training' task is the most complicated one, and to characterize this learning dynamic rather than the noise is of the most interest. Therefore, even if the decomposition is approximately correct for deep learning tasks, it could be such a case that how to characterize the 'bias training' dynamic is still unknown, while the 'variance training' dynamic is not important at all.



**Summary Of The Paper:**

This paper proposes a new approach to characterize the excess risk by decomposing the risk into two parts, the risk of learning the exact response and the risk of fitting the noise signal. Under the linear regression setting, the learning dynamic can be decomposed exactly into the proposed two parts. Stability and uniform convergence arguments are applied to two parts respectively. Similar results are shown under the more general setting, though additional assumptions are made to guarantee a reasonable dynamic decomposition.

**Summary Of The Review:**

The decomposition proposed in this paper makes sense in some specific settings like linear regression. However, it is very unclear for me to see how this decomposition can be extended to a more general regime and how it helps understand the learning dynamics of deep learning. There are also no new technical tools developed. Given that I believe the decomposition's applicability and significance are questionable, and the technical contribution seems incremental, I recommend a weak reject.

---

> ### Author Response · Authors · 2021-11-16
> **Clarifications**
>
> Thank you for your helpful comments! We have found your comments very constructive! Below, we do our best to address your questions, but before that, we encourage you to read the “General Response” section first.
>
> > The first concern is, how better is Theorem 1 compared to directly using uniform convergence to characterize the excess risk of linear regression, especially under the high signal-noise-ration scenario. [...] Some added discussion may help.
>
> Thanks for allowing us to address your concern and provide added discussion. We refer to the part [General Response: Comparison with original stability-based bounds] for a thorough derivation. In summary, our bound outperforms the original stability-based bounds with a large signal-to-noise ratio at a large time $t$.
>
> > The next concern is the applicability of this decomposition. [...] some very strong assumptions are made [...]
>
> The decomposition theorem provided in the current edition is not the only approach to achieving the decomposition goal, as argued in the main text. For example, we conduct several experiments on neural networks to illustrate that neural networks exhibit the decomposition phenomenon, despite the fact that the assumptions do not explicitly hold. We leave the theoretical discussion of the decomposition theorem for future work.
> From the theoretical perspective, we show that, at the very least, the assumptions stated in this section hold in the overparameterized linear regression and diagonal matrix recovery regimes.
>
> > I also have concerns with whether this decomposition is meaningful for modern deep learning tasks. [...] In modern deep learning tasks, usually, the 'bias training' task is the most complicated one [...] while the 'variance training' dynamic is not important at all.
>
> This is a thought-provoking question.
>
> In fact, we switch from standard training under noisy response to variance/bias training, which significantly reduces the generalization challenges since variance/bias training is only a special case of standard training.
> For example, uniform convergence fails in the general overparameterized linear regression (standard training) but works in its noiseless version (bias training) [1, 2].
>
> Besides, variance training is usually more complicated and crucial in generalization analysis (noisy response regimes) since fitting noise usually does harm to the generalization gap (and is usually harder to analyze, see [3]). Therefore, it is improper to say “variance training dynamic is not important at all”. In this paper, we tackle the variance training by the property that models including neural networks usually converge slower when fitting noise.
>
> We finally remark that noisy response is pretty general in practice and widely studied in previous works [4, 5, 6]. Therefore, the analysis under such regimes is pretty valuable.

---

> > ### Author Response · Authors · 2021-11-16
> > **Clarifications (2)**
> >
> > > The decomposition proposed in this paper makes sense in some specific settings like linear regression [...] and the technical contribution seems incremental [...]
> >
> > We apologize for any point of the confusion which makes you feel contribution is incremental since "incremental" is pretty harsh criticism.
> >
> > Firstly, to our best knowledge, this is the first paper to take advantage of the inherited property of different techniques to derive a generalization bound.
> > We point out that different techniques may be effective in different regimes. For example, stability-based bounds are helpful in pure noise settings, and uniform convergence is helpful in noiseless settings.
> > Therefore, the analysis in this paper brings some new insights into the generalization analysis and motivates us to exploit the inherited characteristics of the existing methodologies.
> >
> > Secondly, the decomposition framework approach is not groundless but motivated by the observations of neural networks.
> > The resulting decomposition bounds naturally agree better with neural networks compared to the original stability-based bounds.
> > We believe that accompanying clarifications will aid in evaluating the contributions of the decomposition framework.
> >
> > Once again, thank you for your insightful review! We look forward to any further discussions that would help your evaluation.
> >
> > [1] Nagarajan, V., & Kolter, J. Z. (2019). Uniform convergence may be unable to explain generalization in deep learning. *Advances in Neural Information Processing Systems*, *32*.
> >
> > [2] Negrea, J., Dziugaite, G. K., & Roy, D. (2020, November). In defense of uniform convergence: Generalization via derandomization with an application to interpolating predictors. In *International Conference on Machine Learning* (pp. 7263-7272). PMLR.
> >
> > [3] Bartlett, P. L., Long, P. M., Lugosi, G., & Tsigler, A. (2020). Benign overfitting in linear regression. *Proceedings of the National Academy of Sciences*, *117*(48), 30063-30070.
> >
> > [4] Karimi, D., Dou, H., Warfield, S. K., & Gholipour, A. (2020). Deep learning with noisy labels: Exploring techniques and remedies in medical image analysis. *Medical Image Analysis*, *65*, 101759.
> >
> > [5] Song, H., Kim, M., Park, D., Shin, Y., & Lee, J. G. (2020). Learning from noisy labels with deep neural networks: A survey. *arXiv preprint arXiv:2007.08199*.
> >
> > [6] Ma, X., Huang, H., Wang, Y., Romano, S., Erfani, S., & Bailey, J. (2020, November). Normalized loss functions for deep learning with noisy labels. In *International Conference on Machine Learning* (pp. 6543-6553). PMLR.

---

> ### Comment · Reviewer_HZ9d · 2021-11-29
> **Post-rebuttal update**
>
> The authors have addressed most of my concerns. But still, the motivation of such a decomposition is not clear to me. It is especially so for the classification task where the noise, if any, is low compared to the signal. Therefore, I will keep my score unchanged.

---

### Official Review · Reviewer_MkVE · 2021-11-01

**Correctness:** 3
**Technical Novelty And Significance:** 3
**Empirical Novelty And Significance:** 3
**Recommendation:** 5
**Confidence:** 2

**Main Review:**

As a non-specialist in this topic, I found the paper hard to follow. While the idea of decomposing the excess risk and combining the best of stability and uniform techniques to produce a better bound is very interesting and clear, the writing is confusing. Moreover, the quantity of typos and confusing sentences (some of which I list below) don't help.

**Questions**

1. The authors should explicit better what is the setting considered in the non-linear regime. While in the linear regime it is very clear what is the model of data and the loss, there is a conceptual jump between Sec. 3 and Sec. 4 which makes it confusing to understand how general are the results in Sec 4. Does theorem 2 holds for any model and any data distribution as long as assumptions 1 and 2 are valid? If yes, could the authors provide an intuition of how much these assumptions are constraining?

2. For the real data experiments, how is the noise component estimated?

**Comments / Suggestions**

- In the second paragraph in the introduction, both shortcomings of stability-based bounds refereed by the authors are due to non-convexity. The first is due to the norm of the gradient, but it is not clear what the second is due to. The authors should precise better to what aspect of non-convexity the second obstruction comes from.

- On the related work, "Bias-variance decomposition": [1] proceeded [2] in the investigation of the bias-variance decomposition for the double descent in random features regression.

- *Besides, recent works (e.g., Jacot et al. (2018)) show that neural networks converge to overparametrized linear models (with respect to the parameters) as the width goes to infinity under some regularity conditions*. The authors could be more precise on what they mean by *regularity* here. As it is, the sentence seems to suggest that almost all infinite-width network is lazy.

- The notation of vectors should be consistent. As it is, vectors with latin letters are denoted with bold characters, while vectors with greek letters (such as the weights $\theta$) are not bold.

- The authors speak about neural networks in the abstract, and present two plots in Fig. 2 and Fig. 3 related to neural networks. However, these networks are never defined or refereed to in the main manuscript. The authors should either detail the setting they are plotting in the main, or leave these results to the appendix.

**Minor typos**

- Abstract: *As alternative approaches, techniques based on stability analyze the training dynamics and drive algorithm-dependent generalization bounds.*. This sentence is confusing.

- Abstract: *[...] (a) stability-based bound*.

- Introduction, first paragraph: *which takes (the) supremum*.

- Introduction, second paragraph: *Stability-based bound(s)* or *(The) Stability-based bound*. Appears three times in this paragraph.

- Introduction, footnote: *In this paper, we refer the signal to the clean data without the output noise, and the noise to the output noise.* This sentence is confusing. Maybe: "we refer to the clean data without the output noise as the signal" would be clearer.

- Introduction, third paragraph: *component(s)*

- Introduction, third paragraph: *on both synthetic and real-world dataset(s)* or *on both (a) synthetic dataset and (a) real-world dataset* (the first option is better).

- Introduction, fourth paragraph: *via (a) bias-variance decomposition*.

- Introduction, contributions: *into (a) variance and bias component*.

- Introduction, contributions: *provides interesting insights into the generalization community.*. into -> for?

- Introduction, related work: *Oymak et al. (2019) applied bias-variance decomposition on (the) Jacobian of neural networks to explain their different performances on clean and noisy data.*

- Section 2, third paragraph: *Let $\mathcal{A}_{t}$ denote the optimization algorithm which takes (the) dataset $\mathcal{D}$ as (an) input and return(s) the trained parameter $\hat{\theta}^{t}$ at time $t$*

- Section 2, fourth paragraph: *Although the minimizer (might not) be unique*

- Section 2, fifth paragraph: spilt -> split

- Section 3, page 5: *The notation $B$ in Theorem 1 denote(s)*

- Section 4, page 6: *the only way to reach the* or *the only way of reaching the*

- Section 4, page 6: *minimizer $\theta^{\star}$ of (the) excess risk*

[1] Stéphane D'Ascoli, Maria Refinetti, Giulio Biroli, Florent Krzakala, Double Trouble in Double Descent: Bias and Variance(s) in the Lazy Regime, arXiv:2003.01054 [cs.LG]

[2] Ben Adlam, Jeffrey Pennington, Understanding Double Descent Requires a Fine-Grained Bias-Variance Decomposition, arXiv:2011.03321 [stat.ML]


**Summary Of The Paper:**

This work proposes a new bound for the excess risk based on a decomposition into a bias and variance term. The authors use a stability-based analysis to bound the bias excess risk, and uniform convergence to bound the variance excess risk. They illustrate their framework in two contexts: overparametrised linear regression with SGD and low-rank diagonal matrix recovery with gradient flow.

**Summary Of The Review:**

The work has some interesting new ideas that, to the best of my knowledge, are original. However, the results are presented in a confusing manner, making it hard to understand the scope of applicability of these results to a non-expert. Therefore, I believe this work could benefit from a rewriting.

---

> ### Author Response · Authors · 2021-11-16
> **Clarifications**
>
> Thanks for your detailed comments, and we are happy that you are interested in our idea.
> Below, we do our best to address your questions adequately.
> We encourage you to read the "General Response" section first and then return here for further discussion.
>
> > The authors should explicit better what is the setting considered in the non-linear regime. [...] Does theorem 2 holds for any model and any data distribution as long as assumptions 1 and 2 are valid? If yes, could the authors provide an intuition of how much these assumptions are constraining?
>
> We apologize for not making all the statements completely clear.
> The decomposition theorem indeed holds as long as assumptions 1 and 2 are valid. We present the overparameterized linear regression regimes in Sec.3 for a thorough procedure of the decomposition and Sec.4 for a more general result.
> For the generality of the assumptions, we theoretically prove that the assumptions hold in both linear and non-linear cases.
> Since the assumptions are challenging to verify in more complex cases, we conduct experiments to empirically illustrate that the decomposition phenomenon (the excess risk in standard training can be split into variance components and bias components) generally appears in neural network regimes.
>
> > For the real data experiments, how is the noise component estimated?
>
> Since both MINIST and CIFAR-10 are standard datasets for image classification, it is reasonable to regard that there is no label noise in these two datasets. Hence, we refer the clean training to directly training over the original dataset. As for the standard training, we add random noise to the true label. We refer to Appendix A.2 for more details.
>
> > Comments / Suggestions.
>
> We sincerely appreciate the reviewer for providing such detailed and constructive comments! We will polish the manuscript based on these helpful comments. Thanks!
>
> > Other minor typos.
>
> We apologize for some typos that have slipped through. We make sure to fix them all for the final version.
>
> Once again, thank you for your detailed review and your confirmation on the novelty! We make sure to polish our manuscript based on your detailed and constructive comments. We look forward to any further discussions that would help your evaluation.

---

### Official Review · Reviewer_Pc9J · 2021-11-05

**Correctness:** 3
**Technical Novelty And Significance:** 3
**Empirical Novelty And Significance:** 2
**Recommendation:** 5
**Confidence:** 3

**Main Review:**

I believe decomposing the excess risk into the variance excess risk and bias excess risk intuitively makes sense, which aligns well with the empirical observation that neural networks converge fast when fitting signal but converge relatively slowly when fitting noise. Especially, the authors first pointed out two flaws in stability-based bounds, i.e.,

- stability-based bound depends heavily on the gradient norm
- stability-based bound usually does not work well under general non-convex regimes.

These two flaws make stability-based bound cannot handle well with neural networks, where the gradient norm at initialization is usually large, and the optimization path is highly-nonconvex. By decomposing the excess risk into BER and VER, the authors can hopefully get a better bound. For VER, they apply the stability-based bound on it, as the gradient norm of variance training is usually smaller than that of standard training at initialization and the optimization of VER is closer to convex. Whereas for BER, they adopt the uniform convergence to bound it.
In addition, the theorems cover both the linear regimes and the non-linear regimes as well as accompanied with some empirical verifications. Even though, I have the following concerns:

- Both the linear regime and diagonal matrix recovery regime are too different from the neural network case, as the optimization of neural networks is highly non-convex. Although the NTK theory indicates that the neural network at the infinite-width limit is equivalent to kernel methods, it is still far away from explaining the generalization/optimization property of practical neural networks. Therefore, I believe the claims in the paper about neural networks are not well-supported. I would suggest the authors apply their framework to analyze, for example, two-layer relu networks.

- It is unclear to me how does the new decomposition lead to improvements over the original stability bound quantitatively, or the improvement seems quite marginal?  For example, on page 6, you mentioned that "the bound in Theorem 1 outperforms the stability-based bound when $\|θ^\star\|^2$ is
large compared to V (namely, large signal-to-noise ratio)". However, there is also a $\|θ^\star\|^2$ in Theorem 1, and both bounds are approximately $\tilde{\mathcal{O}}(1/\sqrt{n})$.  I believe a more concrete comparison between your bound and the original stability bound is necessary (e.g., a bound depends on the signal-to-noise ratio?). Otherwise, I am not convinced that your decomposition will indeed improve the original bound.


- The following paper is very relevant to your work, it would be great if you can discuss it in the related work.
-- The Deep Bootstrap: Good Online Learners are Good Offline Generalizers, Preetum Nakkiran, Behnam Neyshabur, Hanie Sedghi International Conference on Learning Representations (ICLR), 2021.

**Summary Of The Paper:**

In this paper, the authors propose to decompose the excess risk into the variance excess risk (VER) and bias excess risk (BER). Based on this decomposition, the authors derive the generalization bound for overparameterized linear regimes and matrix recovery regimes.

**Summary Of The Review:**

Overall, the idea in the paper is well-motivated. However, the results in the paper are mostly about overparameterized linear regression and matrix recovery, but the many of the claims in the paper are about understanding the generalization of neural networks. Besides, I am not convinced that the derived bounds in the paper really improve over the original bounds. Therefore, I am currently leaning towards a weak rejection. If the authors can address my concerns, I will consider increasing my rating.

---

> ### Author Response · Authors · 2021-11-16
> **Clarifications**
>
> We are delighted that you think our idea is well-motivated and novel.
> Below, we do our best to address your questions adequately.
> We encourage you to read the "General Response" section first and return here for further discussion.
>
> > [...] Although the NTK theory indicates that the neural network at the infinite-width limit is equivalent to kernel methods, it is still far away from explaining the generalization/optimization property of practical neural networks. Therefore, I believe the claims in the paper about neural networks are not well-supported. I would suggest the authors apply their framework to analyze, for example, two-layer relu networks.
>
> Thank you for the suggestion on applying the decomposition framework to the general neural network regimes. The things can be divided into two folds. On the one hand, considering NTK regimes, we argue that the decomposition framework can be applied under NTK regimes due to the success of overparameterized linear regression regimes [1]. However, on the other hand, when considering non-NTK regimes (e.g., general two-layer ReLU networks), existing approaches hardly work well, including uniform convergence and stability-based bounds [2].
>
> Unfortunately, most existing analyses on two-layer neural networks are still far from reality, e.g., some studies need to assume the activation does not change much, just like the NTK framework [3]. Therefore, it is unfair and unrealistic to expect the decomposition framework to fully explain the two-layer networks. Instead, the decomposition framework indeed moves forward to neural network regimes compared to the stability-based bounds.
>
> > […] I believe a more concrete comparison between your bound and the original stability bound is necessary (e.g., a bound depends on the signal-to-noise ratio?). Otherwise, I am not convinced that your decomposition will indeed improve the original bound.
>
> We thank the reviewer for the helpful advice. We present a concrete example to demonstrate the improvement in [General Response: Comparison with original stability-based bounds]. Additionally, we emphasize that the most intriguing aspect of the decomposition framework is that it accords better with the deep learning regimes than the numerical improvements on the overparameterized linear regression.
>
> > The following paper (Nakkiran et al., (2021)) is very relevant to your work, it would be great if you can discuss it in the related work.
>
> We thank the reviewer for the recommendation. Nakkiran et al. (2021) take advantage of similar observations (informally, networks learn faster on the population loss) and explain generalization from an online perspective. We will discuss it in the next edition.
>
> > [...] the many of the claims in the paper are about understanding the generalization of neural networks.
>
> This paper indeed makes progress in explaining the generalization of neural networks in the sense that the decomposition framework is based on the properties of neural networks. We refer to [General Response: Contribution of the paper] for more details.
>
> Once again, we appreciate the reviewer's precious time.
> In response to your requested clarifications, we trust that the detailed replies provided to the generality and improvements made above will shed light on the scope and applicability of the contributions.
> We are eager to engage in further discussions to clear out any confusion.
>
> [1] Arora, S., Du, S., Hu, W., Li, Z., & Wang, R. (2019, May). Fine-grained analysis of optimization and generalization for overparameterized two-layer neural networks. In *International Conference on Machine Learning* (pp. 322-332). PMLR.
>
> [2] Nagarajan, V., & Kolter, J. Z. (2019). Uniform convergence may be unable to explain generalization in deep learning. *Advances in Neural Information Processing Systems*, *32*.
>
> [3] Shah, H., Tamuly, K., Raghunathan, A., Jain, P., & Netrapalli, P. (2020). The pitfalls of simplicity bias in neural networks. *arXiv preprint arXiv:2006.07710*.

---

### Author Response · Authors · 2021-11-16
**General Response (1)**

We appreciate the reviewers for their constructive comments that help to enhance our manuscript considerably.
Furthermore, we would like to express our gratitude to the reviewers for their positive feedback on the paper's novelty, which has been a great source of inspiration for us.
Below, we first focus on the comments raised by most of the reviewers and then address each reviewer individually.

- **Contribution of the paper.**

***The merits of the decomposition framework are more conceptual than technological***. That is to say: the decomposition framework naturally has more chance to be applied into deep learning regimes than previous approaches (e.g., uniform convergence [1], stability-based bound [2]).

In this paper, the decomposition framework is inspired by the observation that neural networks converge faster when fitting signals [3, 4].
Empirically, we demonstrate that neural network training indeed can be decomposed into signal and noise components.
Theoretically, we verify two cases (both linear and non-linear regimes) in the current version instead of the general neural networks due to the limits of mathematical tools (and therefore, the improvements of bounds in these cases are just byproducts).

Besides, we emphasize that the decomposition framework takes a step towards understanding neural networks by reducing the generalization challenge of analyzing the noisy response to analyzing the pure noise and the pure signal.
Our ultimate goal, of course, is to understand neural networks.
Although the decomposition framework, like much of the existing literature [5, 6], cannot be directly extended to general neural networks, it still sheds light on uncovering the generalization mysteries of neural networks.

We finally briefly mention the relationship between the general neural network and the two models studied in our paper. First, as the width of the neural network increases, it will gradually converge to the linear regime, which ensures the applicability of our framework since it naturally shares the decomposition property. Second, the factorized model applied in the matrix recovery problem can be considered as a two-layer neural network with quadratic activation, as shown in [8]. Besides, Figure 3 and Figure 4 illustrate the utility of the decomposition framework in general neural networks.

(To Be Continued)

---

> ### Author Response · Authors · 2021-11-16
> **General Response (2)**
>
> - **Comparison with the original stability-based bounds.**
>
> One of the shortcomings of the stability-based bound is its failure with a large time $t$ [2]. Our decomposition framework can aid in reducing the failure at a large time $t$. For example, when considering a large time $T = n^{3/4}$ with $\lambda = 1$, the decomposition bound is $\tilde{\mathcal{O}}(n^{-1/4} [V+B]^2 )$. As a comparison, the original bound is $\tilde{\mathcal{O}}(n^{-1/4} [V+B^\prime]^2 )$. We next show that $B< B^\prime$ holds with a large signal-to-noise ratio.
>
> We write the above formula as $\Vert \hat{\theta}^{(t)}_v \Vert = B <B^\prime =\Vert \hat{\theta}^{(t)} \Vert $.
> From the fomulation of the trained parameter, we have $\hat{\theta}^{(t)}_v = [I - (I - \frac{\lambda}{n} X^{\top} X)^t] X^{+} E$ and $\hat{\theta}^{(t)} = [I - (I - \frac{\lambda}{n} X^{\top} X)^t] X^{+} Y = [I - (I - \frac{\lambda}{n} X^{\top} X)^t] \theta^*  +  \hat{\theta}^{(t)}_v$, where $Y = X\theta^{*} + E$ and $X^{+}$ denotes the pseudo-inverse of $X$.
> Therefore, it suffices to show that $\Vert  [I - (I - \frac{\lambda}{n} X^{\top} X)^t] \theta^* \Vert > 2 \Vert \hat{\theta}^{(t)}_v \Vert $.
>
> Note that the first part is only related to signal $\theta^*$ and the second part is related to noise $E$ (by applying concentration, we can also say the second part is closely related to the noise level $\sigma^2$).
> Therefore, with a large signal-to-noise ratio (informally), the last equation holds. To summary, $B<B^\prime$ holds.
>
> We make sure to add a detailed discussion in the next version.
>
> **Additional Note**: (a) Our proposed bound is better than the original stability-based bound at a large time $t$; (b) To move towards explaining neural networks, we still suggest new frameworks based on the properties of neural networks, as the decomposition framework does.
>
> [1] Nagarajan, V., & Kolter, J. Z. (2019). Uniform convergence may be unable to explain generalization in deep learning. *Advances in Neural Information Processing Systems*, *32*.
>
> [2] Bousquet, O., & Elisseeff, A. (2002). Stability and generalization. *The Journal of Machine Learning Research*, *2*, 499-526.
>
> [3] Zhang, C., Bengio, S., Hardt, M., Recht, B., & Vinyals, O. (2021). Understanding deep learning (still) requires rethinking generalization. *Communications of the ACM*, *64*(3), 107-115.
>
> [4] Arora, S., Du, S., Hu, W., Li, Z., & Wang, R. (2019, May). Fine-grained analysis of optimization and generalization for overparameterized two-layer neural networks. In *International Conference on Machine Learning* (pp. 322-332). PMLR.
>
> [5] Bartlett, P. L., Long, P. M., Lugosi, G., & Tsigler, A. (2020). Benign overfitting in linear regression. *Proceedings of the National Academy of Sciences*, *117*(48), 30063-30070.
>
> [6] Negrea, J., Dziugaite, G. K., & Roy, D. (2020, November). In defense of uniform convergence: Generalization via derandomization with an application to interpolating predictors. In *International Conference on Machine Learning* (pp. 7263-7272). PMLR.
>
> [7] Hardt, M., Recht, B., & Singer, Y. (2016, June). Train faster, generalize better: Stability of stochastic gradient descent. In *International Conference on Machine Learning* (pp. 1225-1234). PMLR.
>
> [8] Li, Yuanzhi, Tengyu Ma, and Hongyang Zhang. "Algorithmic regularization in over-parameterized matrix sensing and neural networks with quadratic activations." *Conference On Learning Theory*. PMLR, 2018.

---

### Decision · Program_Chairs · 2022-01-20

**Decision:**

Accept (Poster)

**Comment:**

The main contribution is a way of analyzing the generalization error of neural nets by breaking it down into bias and variance components, and using separate principles to analyze each of the two components. The submission first proves rigorous generalization bounds for overparameterized linear regression (motivated in a general sense by the NTK); there are settings where this improves upon existing bounds. It extends the case to a matrix recovery model, showing that it's not limited to the linear regime. Finally, experimental results show that the risk decomposition holds empirically for neural nets.

The numerical scores would place this paper slightly below the cutoff. The reviewers feel that the paper is well written and have not identified anything that looks like a critical flaw. They have a variety of concerns, mostly centered around whether the results apply to practical situations. Specifically, they're worried about (1) the theory not applying directly to neural nets, (2) the high-noise setting being less relevant for modern deep learning, and (3) whether there's a realistic situation where it improves over past bounds. Regarding (1), the theory covers not only the linear regime, but also the nonlinear matrix recovery regime; combined with the empirical results, this seems pretty solid by the standards of a DL theory paper. Regarding (2), even though the most common benchmarks indeed have low label noise, the high-noise regime still seems worth understanding (after all, we'd like our nets to work in domains like medicine). I haven't dug deeply enough to properly evaluate (3), but the author response seems believable to me.

Overall, the paper strikes me as creative and well-executed. Regardless of whether the theory is easily extendable to neural nets, this seems like an interesting paper that can be built on in future work. I recommend acceptance.